

# Topothermohaline convection – from synthetic simulations to reveal processes in a thick geothermal system

Attila Galsa[1,2], Márk Szijártó[1,3], Ádám Tóth[4], Judit Mádl-Szőnyi[3]

[1]Department of Geophysics and Space Science, Institute of Geography and Earth Sciences, ELTE Eötvös Loránd University, Budapest 1117, Hungary
[2]Institute of Earth Physics and Space Science, HUN-REN, Sopron 9400, Hungary
[3]József and Erzsébet Tóth Endowed Hydrogeology Chair, Department of Geology, Institute of Geography and Earth Sciences, ELTE Eötvös Loránd University, Budapest 1117, Hungary
[4]Copernicus Institute of Sustainable Development, Utrecht University, Utrecht 3584, The Netherlands

*Correspondence to*: Ádám Tóth (a.z.toth@uu.nl)

**Abstract.** Water table topography, temperature and solute content of groundwater all influence regional groundwater flow. Two-dimensional synthetic numerical calculations were performed to investigate the dynamic interaction between topography-driven forced convection and buoyancy-controlled free thermohaline convection. In the combined topothermohaline model, the recharge and flow-through zones are dominated primarily by topography-driven regional groundwater flow, which drifts warm upwellings towards the discharge zone. Beneath the discharge zone, a dome with high temperature, salinity and water age is formed, in which time-dependent thermohaline convection develops. It was established that (1) increasing the water table gradient suppresses the thermohaline dome, resulting in a near steady-state solution. (2) Increasing the bottom heat flux strengthens the warm upwellings, which ultimately leads to the break-up of the thermohaline dome, thus, paradoxically, reducing the average temperature. (3) Increasing the bottom salt concentration weakens the topography-driven groundwater flow leading to the formation of a multilayered thermohaline dome with extremely high temperature, salinity and age. The operation of the topothermohaline model was demonstrated along a hydrogeological section crossing the Buda Thermal Karst (BTK), Hungary. We found that the unconfined karstic areas are dominated by topography-driven water flow, while in the confined, deep reservoirs, thermohaline convection is the prevailing flow regime. The thermally and compositionally mixed water promotes karstification and reaches the surface near the Danube River, the main discharge area. In the eastern, confined areas of BTK, significant amounts of heat may be retained on a geological time scale, making it a promising site for geothermal exploration.

## 1 Introduction

Deep and thick aquifers in sedimentary basins contain significant thermal water and mineral resources. Although limited knowledge of their physical flow processes exists, their exploitation is becoming increasingly needed worldwide. Fluid migration in basins is a key factor in the large-scale mobilization of heat and matter (Tóth, 1999; Ingebritsen et al., 2006; Bredehoeft 2018). Thus, revealing basins' flow and transport processes can help improve the success of exploiting targeted



resources (Czauner et al., 2022). These topics are especially significant because geothermal installations, new technologies for critical mineral production, and $CO_2$ sequestration use these aquifers for both fluid production and injection.

The eternal dilemma: What drives groundwater flow in sedimentary basins? The first mathematical answer was given
by József Tóth, who revealed that horizontal variations in the water table can cause groundwater movement, especially in small (1–2 km deep) drainage basins. At least, he came to this conclusion through his simple, two-dimensional, homogeneous and isotropic analytical models (Tóth, 1962; 1963), which was later improved by Robinson and Love (2013). The water table often follows the topography of the surface and can be considered as a subdued replica thereof — depending on climate, the slope of the surface, (hydro)geological conditions, etc (Haitjema and Mitchell-Bruker, 2005) —, resulting in the so-called
topography-driven groundwater flow (Tóth, 2009; Bresciani et al., 2016). Since the theory became widespread, several analytical studies have been carried out to make this simple but mathematically robust model more realistic (e.g. Freeze and Witherspoon, 1967), taking into account, for example, the depth dependence of permeability or permeability anisotropy (e.g. Jiang et al., 2009; 2011; Wang et al., 2011). The concept has achieved even greater success in numerical modelling, and has been able to explain the nature of groundwater flow in numerous cases (Winter and Pfannkuch, 1984; Gleeson and Manning,
2008), and its relationship with wetland distribution (Hayashi et al., 2016; Tóth et al., 2016; Simon et al., 2023), changes in groundwater-dependent ecosystems (Havril et al., 2018; Szabó et al., 2023; Fabiani et al. 2024), the complexity of groundwater flow system influenced by climate change (Trásy-Havril et al., 2022; Tsypin et al., 2024) the formation of dynamic hydrocarbon traps (Czauner and Mádl-Szőnyi, 2011; Wendebourg et al., 2018), the appearance of near-surface temperature anomalies, hot springs and flowing wells (An et al., 2015; Mádl-Szőnyi and Tóth, 2015; 2017; Jiang et al. 2020; Tóth et al.,
2020; 2023). Topography-driven conceptual models are successful primarily in shallow depth ranges (Ferguson et al., 2023) where the role of other driving forces could generally be negligible.

As temperature increases, the density of water decreases (at least above 4 °C), and since in most geological environments, temperature increases with depth, thermal buoyancy develops in deeper parts of the groundwater basins. If the positive thermal buoyancy is sufficiently large, in other words, if the Rayleigh number of the system exceeds a critical value,
free thermal convection occurs in both homogeneous and isotropic (Lapwood, 1948; Wooding, 1957; Elder, 1967) and inhomogeneous (Rubin, 1981) porous media. The onset and nature of convective flow is influenced by several factors, such as the Rayleigh number for the layer (Graham and Steen, 1994; Otero et al., 2004; Hewitt et al., 2014), the anisotropy of permeability (Smith and Chapman, 1983), the slope of the water table (Cserepes and Lenkey, 2004) etc. However, in terrestrial regions, the role of topography-driven water flow can rarely be ignored (Mádl-Szőnyi et al., 2023). One of these few cases is
the phenomenon of deep confined geothermal reservoirs e.g. in the Luttelgeest fractured carbonate platform, the Netherlands (Lipsey et al., 2016), in the fissured and karstic carbonate aquifer of Hainaut, Belgium (Licour, 2014), in the overpressured and fractured Mesozoic carbonate formation of Fábiánsebestyén, Hungary (Galsa et al., 2022a) etc.

Szijártó et al. (2019) performed a systematic simulation set in synthetic model domains to reveal the interaction between topography-driven forced and buoyancy-driven free thermal convection. They found that while the increasing layer
thickness and temperature difference across the layer promote the dominance of free thermal convection, it is inhibited by the





high gradient of the groundwater table and the permeability anisotropy. Since variations in the water table almost always influence groundwater flow in a real hydrogeological environment, most case studies — which do not neglect the effect of temperature-dependence of water density — treat these two factors influencing groundwater flow together. Clauser and Villinger (1990) modelled the heat transport process in the sedimentary basin of Rheingraben, Germany, finding that free

thermal convection increases heat transport, even if its presence is detectable only in certain parts of the model domain. 3D numerical model of heat transport in the Northeast German Basin suggests that topography-driven forced convection is the dominant heat transport process in shallow domains, while conduction is the dominant factor in deeper zones (Kaiser et al., 2011; 2013; Noack et al., 2013). However, they demonstrate that free thermal convection also affects heat transport locally. Lopez et al. (2016) explain the temperature and hydraulic head anomalies observed in the Lake Chad Basin by the phenomenon

of mixed (i.e. combined forced and free) thermal convection developing in thick aquifers and being connected through faults. 3D but stationary model of the coupled topography-driven forced and buoyancy-driven free thermal convection is consistent with historical hydrogeological records and thermal anomalies observed in the Independence Basin, Mexico (Ortega Guerrero, 2022).

       The solute content of groundwater varies over a wide range, from practically solute-free rainwater (<0.05 g l$^{-1}$) to

sink hole brines near the Dead Sea (519 g l$^{-1}$) (Weisbrod et al., 2016). Thus, total dissolved solids (TDS) also affect water density, with higher TDS resulting in higher water density, which creates a downward haline buoyancy. The problem of free haline convection and its analytical/numerical solution is very similar to the problem of free thermal convection (Weatherill et al., 2004; Voss et al., 2010). Understanding the groundwater flow controlled by both topography and haline buoyancy is of high importance to coastal populations, as the increasing number and yield of producing wells facilitate subsurface seawater

intrusion and salinization of freshwater aquifers (e.g. Barlow and Reichard, 2010; Hussain et al., 2019). In the interior of continents, the overproduction of water wells for agriculture, drinking water supply and energy use can lead to the mixing of different salinities and salinisation of soils (Yu et al., 2024). Numerical modelling of coupled topography-driven forced and salinity-driven free convection revealed the complex flow regime beneath playas in the Great Basin, Utah, USA (Duffy and Al-Hassan, 1988), elucidated the variability of submarine groundwater discharge in a heterogeneous coastal volcanic aquifer

near the Big Island of Hawaii, USA (Kreyns et al., 2020), characterized the mixed convection below a saline disposal basin depending on the haline Rayleigh number, the heterogeneity and anisotropy of the aquifer (Simmons and Narayan, 1997), determined the effect of saline water intrusion on the pattern of synthetic nested groundwater flow system (Zhang et al., 2020). Galsa et al. (2022b) accomplished a sensitivity test to investigate the influence of salinity, permeability, anisotropy, heterogeneity, dispersivity and water table configuration on the groundwater flow and solute concentration. The operation of

the mixed, so-called topohaline convection model was demonstrated in a realistic hydrogeological environment to explain the connection between the upper siliciclastic and the deep karstic carbonate aquifers through the faulted clayey aquitard with high TDS.

       In thermohaline convection, the free positive thermal and negative haline convection play a competitive role, when both the temperature and the salinity increase with depth. The onset of thermohaline convection was determined analytically



in simple homogeneous Boussinesq models for different boundary conditions by Nield (1968), and in heterogeneous (layered)
aquifers by Rubin (1982a; 1982b). Szijártó and Galsa (2020) studied the stationary and time-dependent character of
thermohaline convection in their two-dimensional synthetic numerical models. In real hydrogeological situations, however,
the effect of forced convection induced by the groundwater table configuration cannot be neglected. Thus, de Hoyos et al.
(2012) investigated the 'thermohaline' effects on 3D groundwater flow forming in the Paris sedimentary Basin, France, but
they concluded that the influence of the pressure and temperature on the water density is insignificant (<3%), so the role of
thermal buoyancy was ignored. Gupta et al. (2015) applied a fully coupled topothermohaline numerical model to determine
the age and origin of brines in the Alberta Basin, Canada. Based on transient simulations, regional, topography-driven
groundwater flow characterizes the shallow formations in the Tiberias Basin, while deep-seated thermohaline convection cells
develop in thick, covered carbonate sequences (Magri et al., 2015).

The Buda Thermal Karst (BTK), Hungary is an area with high geothermal potential, but also a complex groundwater
flow and thermal regime owing to the fact that the karstified carbonates are mostly unconfined on the western side and confined
on the eastern side (Mádl-Szőnyi and Tóth, 2017). Therefore, Szijártó et al. (2021) carried out an extensive numerical study to
clarify the role of different heat transport processes, separating the phenomena of conduction, forced and free thermal
convection. Simulations showed that different heat transport processes dominate in distinct regions of the karst system, while
free thermal convection, developed mainly in deeper and confined reservoirs, formed in all model calculations. Havril et al.
(2016) studied the flow and heat transport processes in the karstified reservoir from the perspective of the geological evolution
of BTK. Based on their numerical model results, it can be stated that with the uplift and erosion of the cover on the western
side, the topography-induced forced convection has been enhanced at the expense of free thermal convection, but the latter is
still operating in the present system, mainly in the deeper, covered carbonates. Although most recent karst waters are freshwater
(rarely brackish), they were formed in a marine environment. Additionally, the clayey cover layer on the eastern side still
contains saline water, which is presumably connected to the water flows in the confined karst (Mádl-Szőnyi et al., 2019). In
such a hydrogeological setting, both topography-driven forced convection and free thermohaline convection can influence the
coupled flow-thermal-saline regime, forming an interconnected geothermal reservoir.

    In all real cases, the terrestrial groundwater flow system is controlled by dynamic interaction of topography-driven
forced convection and density-driven free thermal and haline convection. Still, in most models, the numerical modeller
switches off one or two components of driving forces to reduce computational demands. Sometimes, this is not a problem,
especially if one component of the system is significantly under-dominated, and consequently, the model can produce solutions
that are close to reality. However, in deep basins, where distinct geological formations have different hydraulic, thermal and
salinity properties, it inevitably leads to oversimplification of the groundwater flow system. Moreover, in such complex, non-
linear systems, small interventions in the model parameters can lead to radically different solutions.

    Therefore, the main goal of this paper is to (1) present an extensive parameter analysis in a two-dimensional synthetic
model domain emphasizing the complex character of the groundwater flow as a dynamic balance between the topography-
driven forced and buoyancy-driven free thermohaline convection. The flow, the temperature and the salinity of the



topothermohaline system are stimulated directly by the water table configuration, the heat flux and salt concentration

prescribed along the bottom of the model domain, respectively. However, the other two variables are also affected in the fully

coupled system. Control parameters, such as the average Darcy flux, the solute concentration, the temperature, and the water

age are computed for both the whole model basin and the different salinity zones to characterize the flow systems

quantitatively. Second, (2) the operation of the topothermohaline convection model is demonstrated in a thick carbonate

system, Buda Thermal Karst (BTK) system, which has a high geothermal potential (Mádlné Szőnyi et al., 2018; Mádl-Szőnyi

et al., 2019; Mádl-Szőnyi, 2019; Kun et al., 2024). This study is the first to systematically investigate the phenomenon of

topothermohaline convection through synthetic numerical modelling and the first attempt to explore the coupled groundwater

flow, thermal and salinity character of BTK, Hungary.

## 2 Model description

### 2.1 Physical background

The topothermohaline problem is treated numerically by solving the coupled continuity equation (1), Darcy's law (2), and the

heat (3) and mass (4) transport equations (Nield and Bejan, 2017), which govern the conservation of mass and momentum,

and the transport of heat and solute concentration in subsurface water flow, respectively,

$$\Phi \frac{\partial \rho_w}{\partial t} + \nabla(\rho_w \mathbf{q}) = 0, \tag{1}$$

$$\mathbf{q} = -\frac{\mathbf{k}}{\mu}(\nabla p + \rho_w g \nabla z), \tag{2}$$

$$[\Phi \rho_w c_{pw} + (1-\Phi)\rho_m c_{pm}]\frac{\partial T}{\partial t} = -\rho_w c_{pw}\mathbf{q}\nabla T + \nabla\{[\Phi \lambda_w + (1-\Phi)\lambda_m]\nabla T\}, \tag{3}$$

$$\Phi \frac{\partial c}{\partial t} = -\mathbf{q}\nabla c + \nabla[(\Phi D_{diff}\mathbf{I} + \mathbf{D_{disp}})\nabla c], \tag{4}$$

where $\mathbf{q}$, $p$, $T$ and $c$ are the unknown Darcy flux, pressure, temperature and solute concentration, $\Phi$, $\mathbf{k}$, $\mu$, $g$, $D_{diff}$ and $\mathbf{D_{disp}}$, $t$, $z$

and $\mathbf{I}$ denote the porosity, the diagonal permeability tensor, the dynamic viscosity of water, the gravitational acceleration, the

molecular diffusion of water, the mechanical dispersion tensor, the time, the vertical coordinate and the identity matrix,

respectively. $\rho$, $c_p$ and $\lambda$ are the density, the specific heat and the thermal conductivity of the water and the matrix denoted by

subscripts $w$ and $m$, respectively. The dispersion tensor is determined by the longitudinal and transverse dispersivity ($\alpha_L$ and

$\alpha_T$), where the value of $\alpha_T$ was fixed as $\alpha_T = \alpha_L/10$. In the synthetic models, dispersivity was assumed to be zero, and the

permeability was isotropic, while in the real hydrogeological simulation, $\alpha_L = 100$ m and anisotropic permeability ($\varepsilon = k_x/k_z > 1$)

were applied. Constant model parameters for the synthetic model are given in Table 1.

The numerical modelling of topothermohaline convection had to take into account the dependence of water density,

$\rho_w(T,p,c)$, on temperature (sixth-order polynomial) and pressure (second-order polynomial) (Magri, 2009), as well as on

concentration (linear) (Kohfahl et al., 2015). Molecular diffusion, $D_{diff}(T)$, is strongly dependent on temperature, which was





considered in the simulations using a quadratic Arrhenius law (Easteal et al., 1989). In order to calculate the age of the groundwater, the mean age was computed by solving the age mass equation defined by Goode (1996). Goode (1996) supposed

that water age propagates through the porous medium in the same way as solute, by advection, diffusion and dispersion,

$$\Phi \frac{\partial \tau}{\partial t} = -\mathbf{q}\nabla\tau + \nabla\big[(\Phi D_{diff}\mathbf{I} + \mathbf{D_{disp}})\nabla\tau\big] + \Phi, \tag{5}$$

where $\tau$ denotes the mean age of the groundwater, and the last additive term in Eq. (5) represents that the water age increases with time. Thus, Eq. (5) was solved by coupling to Eq. (1)–(4), where water age is influenced by the Darcy flux through the terms of advection and dispersion, and by the temperature through molecular diffusion, $D_{diff}(T)$. Certainly, water age has no

effect on thermohaline convection.

**Table 1. Parameters of the synthetic model**

| Description | Symbol | Value | Unit |
|---|---|---|---|
| Porosity | $\Phi$ | 0.1 | 1 |
| Permeability | $k$ | $10^{-13}$ | m$^2$ |
| Gravitational acceleration | $g$ | 9.81 | m s$^{-2}$ |
| Reference water density | $\rho_{ref}$ | 999.79 | kg m$^{-3}$ |
| Dynamic viscosity of the water | $\mu$ | $5 \cdot 10^{-4}$ | Pa s |
| Matrix density | $\rho_m$ | 2450 | kg m$^{-3}$ |
| Specific heat of water | $c_{pw}$ | 4200 | J kg$^{-1}$ K$^{-1}$ |
| Specific heat of matrix | $c_{pm}$ | 900 | J kg$^{-1}$ K$^{-1}$ |
| Thermal conductivity of water | $\lambda_w$ | 0.6 | W m$^{-1}$ K$^{-1}$ |
| Thermal conductivity of matrix | $\lambda_m$ | 2 | W m$^{-1}$ K$^{-1}$ |
| Longitudinal dispersivity | $\alpha_L$ | 0 | m |
| Transverse dispersivity | $\alpha_T$ | 0 | m |

## 2.2 Numerical model

Synthetic model simulations were carried out in two-dimensional, simple, homogenous, isotropic model domain without

dispersion to concentrate on the hydrogeophysical behaviour of the system. The length and the average depth of the model domain are $L$=40 km and $d$=5 km, respectively (Fig. 1). Boundary conditions for the flow are no-flow along the side walls and the bottom, the regional groundwater flow is driven by a cosinusoidal water table (e.g. Domenico and Palciauskas, 1973; Wang et al., 2015; Zhang et al., 2018) with an amplitude of $H$=50 m in the base model,

$$z_{wt}(x) = -H \cos\left(\frac{\pi x}{L}\right), \tag{6}$$





where $x$ denotes the horizontal coordinate. The water table, $z_{wt}$, corresponds to the top of the model domain, resulting in a topography-driven groundwater flow from right to left (Fig. 1.a). The surface boundary condition for the water age is an open boundary representing that the age of recharging water is zero and the 'age mass' discharges from the model with the water,

$$\tau_s = \begin{cases} 0 & \text{if } \mathbf{nq} < 0 \quad (recharge) \\ \mathbf{n}D_{diff}\nabla\tau = 0 & \text{if } \mathbf{nq} > 0 \quad (discharge) \end{cases}, \tag{7}$$

where $\mathbf{n}$ is the normal vector of the surface. Other boundary conditions are summarized in Fig. 1.


Figure 1. Initial and boundary conditions in the synthetic base model for (a) the flow, (b) the temperature, (c) the salt concentration and (d) the groundwater age. (a) White streamlines and red arrows illustrate the direction of topography-driven groundwater flow.





The initial condition for the Darcy flux is obtained from the solution of Eqs (1)–(2) using reference water density.
Initial conditions for the temperature and the salt concentration come from conductive and diffusive solutions of heat (Eq. (3))
and mass (Eq. (4)) transport equations, respectively. Initially, the water age was zero over the entire model domain (Fig. 1).

The topothermohaline convection model was solved numerically using COMSOL Multiphysics 5.3a, a finite element
software package (Zimmerman, 2006). The synthetic model domain was discretized by approx. 57 000 triangular finite
elements depending on the groundwater table amplitude, $H$. Maximum size of finite elements was 100 m, but boundary
elements with a thickness of 100 m/8 were applied along the boundaries to consider high gradients of the variables. Pressure
and Darcy flux were approximated by quadratic functions, while temperature, concentration and age were assumed to be linear
within the finite elements. Time stepping was increased exponentially to 1000 yr, then it was linear with a maximum time step
of 100 yr. Calculations were carried out to reach the quasi-stationary solution, which required 2–10 Myr depending on $H$. In
general, higher water table amplitude stabilized the system and resulted in a faster quasi-stationary solution. Using this
parameter setup, one simulation consumed approx. 5 GB memory and 30–540 hours CPU time on an Intel Server with 2 Xeon
Gold processors. Overall, 29 time-dependent synthetic numerical simulations were performed to reveal the character of
topothermohaline convection in basin-scale groundwater flow. The numerical model built in COMSOL Multiphysics has been
tested in many hydrogeological situations (for heat/solute/age transport calculations) and has been quantitatively verified (e.g.
Szijártó et al., 2019; Galsa et al., 2022b; Szijártó et al., 2024).

**2.2 Model parameterization**

In the synthetic simulation set, three parameters, the water table amplitude, $H$, the bottom heat flux, $q_{Tb}$ and the bottom salt
concentration, $c_b$, were varied systematically to investigate their direct influence on the Darcy flux, the temperature and the
salinity, respectively. Clearly, in a coupled system, e.g. increasing $H$ not only affects the flow but also indirectly influences
the other parameters. The parameters of the base model and the parameter intervals applied in synthetic simulations are
summarized in Table 2. The parameter ranges were chosen to include the most common examples of groundwater basins, as
well as the case of the Buda Thermal Karst.

The effects of model parameters were investigated and will be presented in three different ways:

- Snapshots of the calculated Darcy flux, temperature, salt concentration, salinity zone and age distribution after the
system reached the steady-state solution illustrate the behaviour of topothermohaline convection.
- The spatially and temporally averaged control parameters, such as mean Darcy flux ($q_{av}$), temperature ($T_{av}$),
concentration ($c_{av}$) and water age ($\tau_{av}$), quantify the role of the model parameters on the complex flow system. The
spatial averaging is not only performed for the full model domain but also for the four salinity zones defined by
Deming (2002) for freshwater (0–1 g l$^{-1}$), brackish water (1–10  g l$^{-1}$), saline water (10–100  g l$^{-1}$) and brine water
(>100  g l$^{-1}$) zones. The relative area of each salinity zone ($A$) is also shown.
- Animations of the variables in the Supplementary material (Galsa et al., 2025) facilitate understanding the dynamics
of topothermohaline convection in synthetic groundwater basins.





**Table 2. Model parameters of the base model and the studied parameter range.**

| Description | Symbol | Base model | Interval | Unit |
|---|---|---|---|---|
| Water table amplitude | $H$ | 50 | 5–500 | m |
| Bottom heat flux | $q_{Tb}$ | 90 | 50–120 | mW m$^{-2}$ |
| Bottom salt concentration | $c_b$ | 70 | 2–200 | g l$^{-1}$ |

## 3 Results

### 3.1 Synthetic simulations

In this chapter, we present the behaviour of the flow/thermal/solute/age mass system in the synthetic base model. Then, the effect of the groundwater table amplitude, the heat flux and salt concentration prescribed at the bottom of the model domain on the control parameters will be systematically investigated.

### 3.1.1 Base model

In the base model, the constant model parameters listed in Table 1 characterize the basin, while the water table amplitude, the bottom heat flux and salt concentration were fixed at $H$=50 m, $q_{Tb}$=90 mW m$^{-2}$ and $c_b$=70 g l$^{-1}$ (Table 2), respectively. Fig. 2 shows the quasi-stationary solution of the Darcy flux, the temperature, the salt concentration and the salinity zones, as well as the water age at $t$=3 Myr after the initial state. Three main zones are separated: (1) the middle part of the basin is dominated by topography-driven regional groundwater flow. This domain is characterized by cool ($T$<20 °C), fresh ($c$<1 g l$^{-1}$) and young ($\tau$<10 kyr) water. Hot and saline plumes are forming at the bottom of the model, and drifting to the left, towards the discharge zone driven by the topography-driven groundwater flow (Fig. 2.b and c). On the other hand, (2) a quasi-stationary thermohaline dome is forming beneath the discharge zone, in which hot ($T$>100 °C), saline ($c$>40 g l$^{-1}$) and aged ($\tau$>400 kyr) water is convecting. At the margin of the two zones, plumes driven by the regional groundwater flow are climbing up the thermohaline dome, resulting in a maximum of Darcy flux (Fig. 2.a). Intense flow is noticed inside the dome, where thermohaline convection is vigorous. (3) The weak thermohaline convection zone beneath the recharge area (right) is the combined consequence of the cold downward flow, the upward thermal buoyancy and the no-flow vertical boundary condition (Szijártó et al 2019). Animation S1 in the Supplementary material (Galsa et al., 2025) illustrates the dynamic balance between the topography-driven forced and thermohaline buoyancy-driven free convection in the base model.





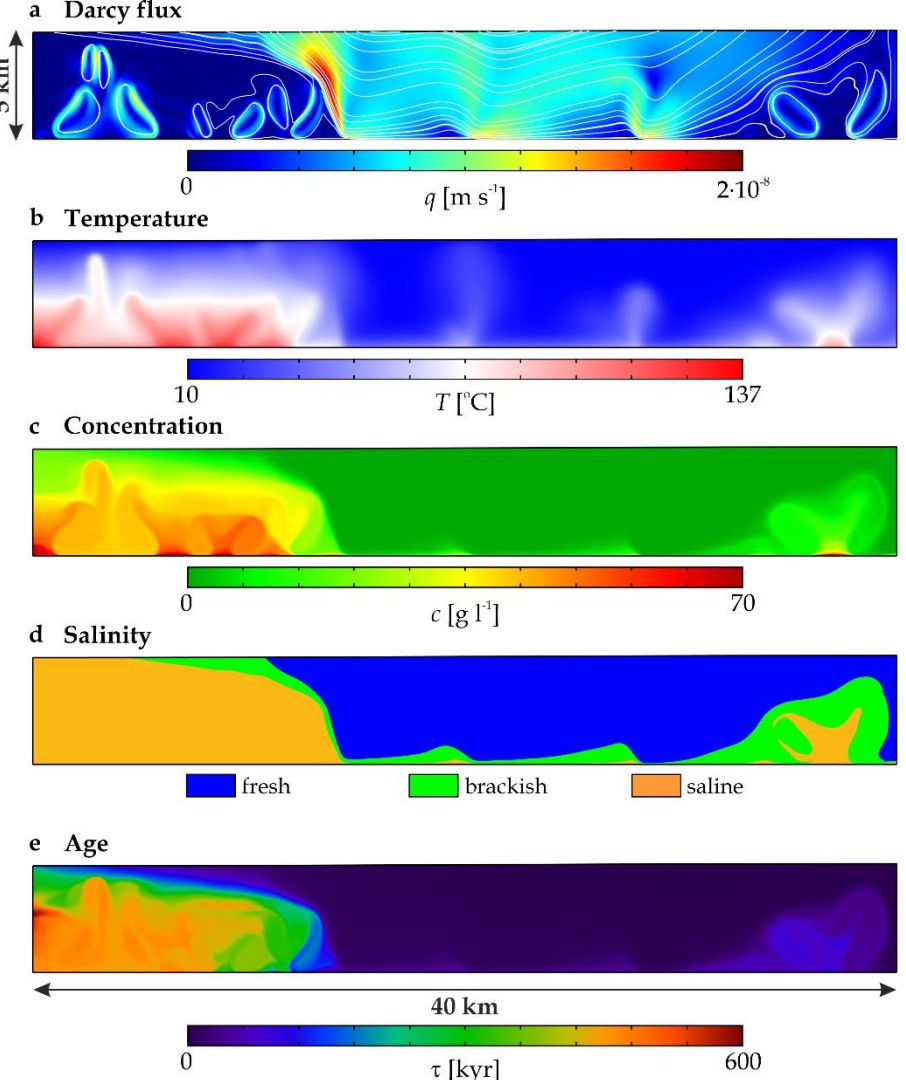

**Figure 2.** Snapshots of the **(a)** Darcy flux, **(b)** temperature, **(c)** salt concentration, **(d)** salinity zones and **(e)** water age at $t$=3 Myr after the initial state in the base model (Table 1 and 2). **(a)** Streamlines illustrate the flow direction. **(d)** Salinity zones: freshwater (blue), brackish water (green) and saline water (orange).

### 3.1.2 Effect of water table amplitude

In this part, only the water table amplitude was varied in the range of $H$=5 m and 500 m, and other model parameters ($q_{Tb}$, $c_b$)

were kept fixed according to the base model (Table 2). The water table amplitude, in other words, the magnitude of the hydraulic gradient, directly influences the intensity of the topography-driven groundwater flow. Figure 3 presents two model calculations having low, $H$=20 m (left) and high, $H$=200 m (right) water table amplitude. At low hydraulic gradients ($H$=20 m), regional water flow is weak to suppress thermohaline convection. Thus, a major part of the basin is saturated by saline, warm and aged water (Fig. 3.b–e, left panel). The topography-driven flow is only detected in a thin band, in the upper right part of



the model domain. Intense free convection occurs in the thermohaline layer (Animation S2 in Galsa et al., 2025). In the absence of a significant regional groundwater flow, the solution is time-dependent and converges only slowly to the quasi-stationary solution. In contrast, at high water table amplitude (*H*=200 m), regional flow dominates the system with a Darcy flux distribution analogous to the solution with constant water density (Fig. 3.a cf. Fig. 1.a). Strong topography-driven forced convection flushes heat and solutes out of the basin, resulting in cold, fresh, young water over a large part of the model domain

(Fig. 3.b–e, right panel). The thermohaline dome shrinks in the deeper part of the discharge zone, leading to a weakly time-dependent numerical solution (Animation S3 in Galsa et al., 2025).

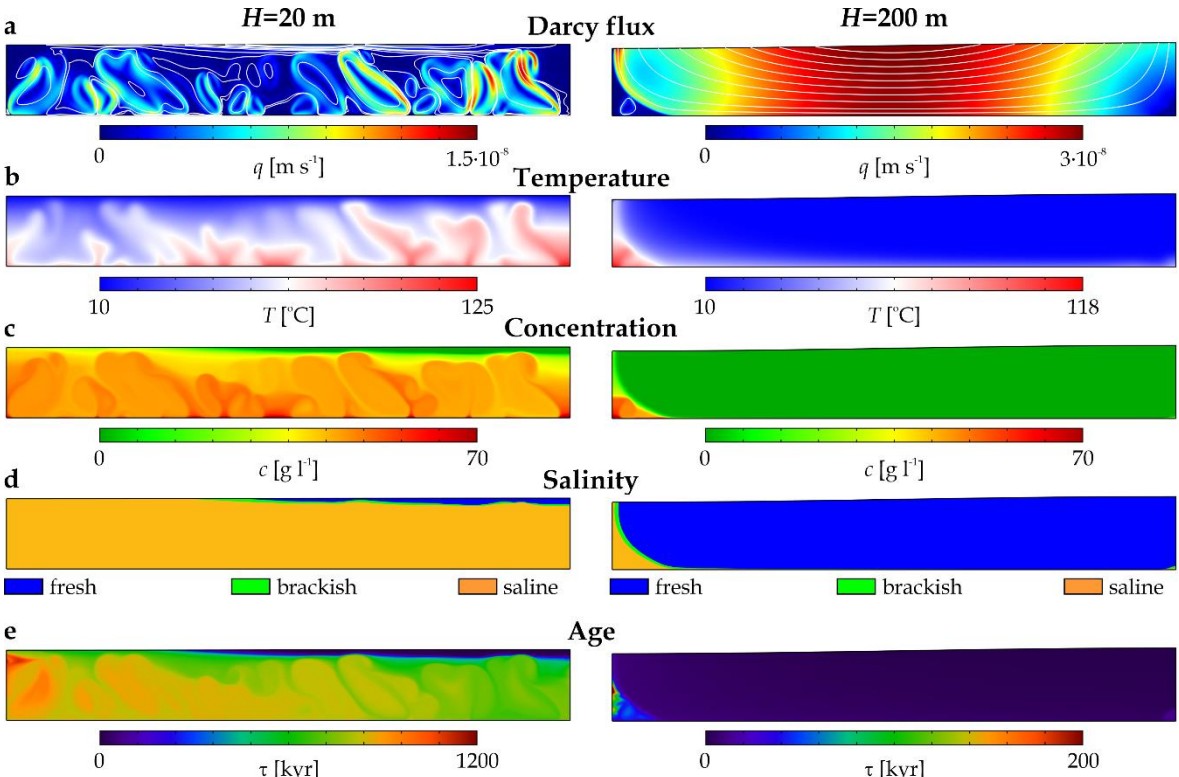

**Figure 3. Numerical solutions of (a) the Darcy flux, (b) the temperature, (c) the salt concentration, (d) the salinity zones and (e) the**
**water age at water table amplitudes of *H*=20 m (left panel) and *H*=200 m (right panel). The quasi-stationary solutions are shown at 10 Myr (left) and 2 Myr (right).**

      Figure 4 quantifies the effect of the water table amplitude on the relative area, Darcy flux, temperature, concentration and water age of the salinity zones and the total model domain. Figure 4.a proves that at low water amplitude (*H*≤20 m), the saline zone controls more than 90% of the basin due to the dominance of thermohaline convection. On the other hand, at high

water table amplitude (*H*≥200 m), the freshwater zone prevails over the majority of the basin (>90%) due to the intense topography-driven forced convection. Between the two extremes, the role of topography-driven groundwater flow becomes more intense as *H* increases. The thermohaline layer that fills most of the basin becomes a dome and shrinks to the deeper



regions of the discharge zone. Higher values of standard deviation indicate that this transition interval ($H$=20–200 m) is strongly time-dependent. In the region dominated by thermohaline convection ($H$≤20 m), the average Darcy flux is equal to
that in the saline zone, $q_{av} \approx 3 \cdot 10^{-9}$ m s$^{-1}$. While the appearance of the freshwater zone intensifies the flow, the average Darcy flux increases approx. linearly following the Darcy's Law (Eq. (2)) (Fig. 4.b). In the thermohaline convection-dominated domain, the average temperature (46–57 °C), salt concentration (45–41 g l$^{-1}$) and water age (981–726 kyr) of the basin are characterised by the saline zone (Fig. 4.c–e). However, forced convection driven by the increasing water table amplitude effectively sweeps heat, salt and aged water out of the basin, resulting in an intense flow of cold, fresh and young groundwater.
It is worth noting that (1) water age and salt concentration are very sensitive to the presence of topography-driven groundwater flow, decreasing by almost three and two orders of magnitude, respectively, with increasing $H$. (2) The temperature, concentration and age of the saline zone exceed the values of the freshwater zone in all simulations.

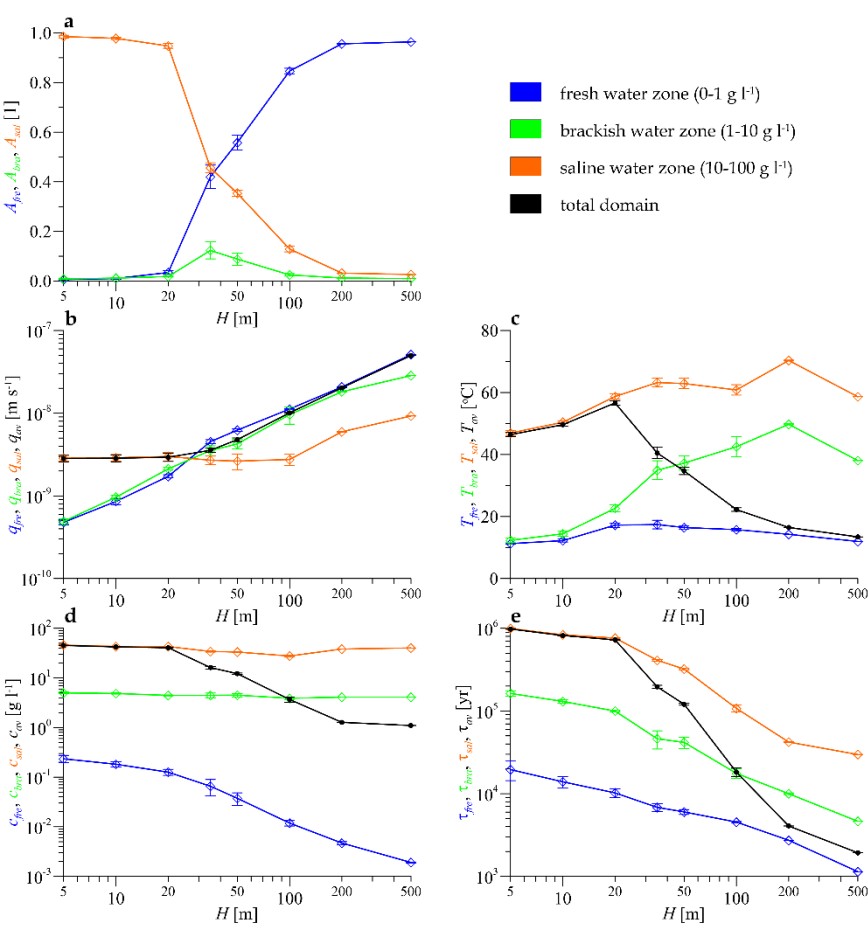

**Figure 4. (a) The relative area, (b) the Darcy flux, (c) the temperature, (d) the concentration and (e) the age of the salinity zones plotted against the water table amplitude. Each salinity zone is marked with a different colour. The values averaged for the whole model domain are shown in black. Standard deviations calculated for the quasi-stationary phase of the time series are marked by error bars.**





### 3.1.3 Effect of bottom heat flux

The heat flux in the continental crust varies over a wide range, which directly influences the temperature distribution and, thus the character of topothermohaline convection. At low bottom heat flux, $q_{Tb}$=50 mW m$^{-2}$ (Fig. 5, left panel), the quasi-stationary solution is similar to the base model (Fig. 2). The model domain in the middle part and below the recharge zone is dominated by topography-driven groundwater flow. While beneath the discharge zone, a thermohaline dome is formed where warm, saline and aged water is convecting (Animation S4 in Galsa et al., 2025). The temperature of the dome is lower than in the

base model due to the lower heat flux. Therefore, the thermohaline convection is more sluggish ($q_{sal}\approx$1.5·10$^{-9}$ m s$^{-1}$), smaller in extent, and contains older water compared to the base model. On the other hand, the high bottom heat flux increases the role of thermal buoyancy and results in strong free thermal convection (Fig. 5, right panel). The warm upwellings evolving at the bottom of the basin drift with the regional water flow towards the discharge zone, resulting in Darcy flux, concentration and age perturbations in the topography-dominated zone. Animation S5 (Galsa et al., 2025) qualitatively suggests that the thermal

buoyancy is so strong that it overcomes the negative haline buoyancy and significantly weakens the thermohaline dome beneath the discharge zone.

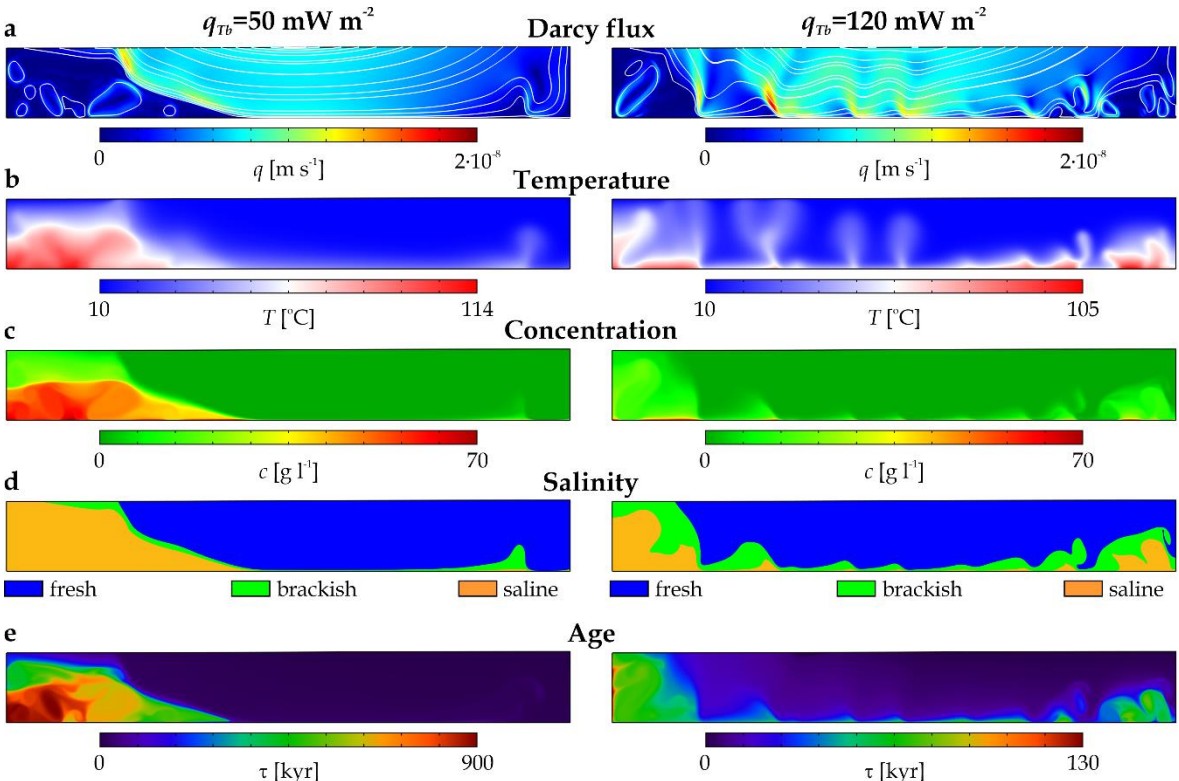

**Figure 5. Numerical solutions of (a) the Darcy flux, (b) the temperature, (c) the salt concentration, (d) the salinity zones and (e) the**
**water age at bottom heat fluxes of $q_{Tb}$=50 mW m$^{-2}$ (left panel) and $q_{Tb}$=120 mW m$^{-2}$ (right panel). The quasi-stationary solutions are shown at 3 Myr (left) and 2 Myr (right).**





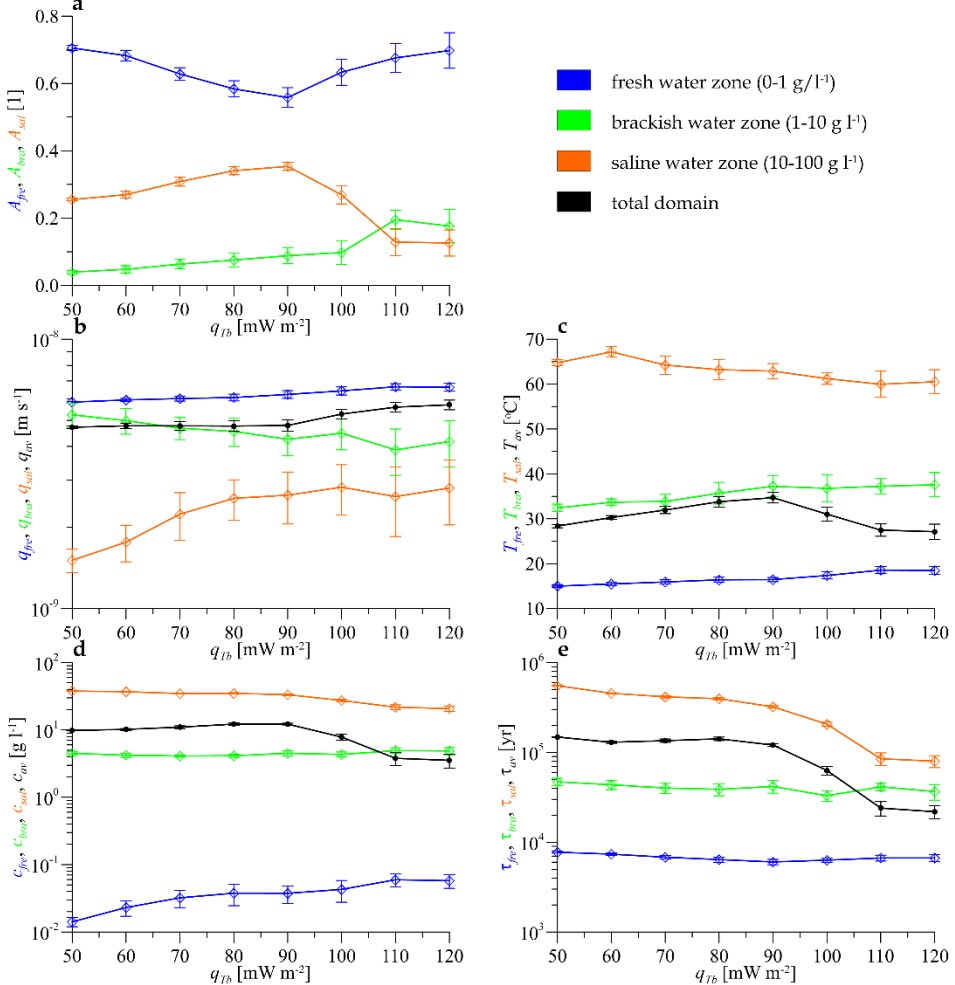

**Figure 6. (a) The relative area, (b) the Darcy flux, (c) the temperature, (d) the concentration and (e) the age of the salinity zones plotted against the bottom heat flux. Each salinity zone is marked with a different colour. The values averaged for the whole model**
**domain are shown in black. Standard deviations calculated for the quasi-stationary phase of the time series are marked by error bars.**

The variation of the control parameters supports the conclusions drawn from the qualitative solutions, that is, the dual role of the bottom heat flux. Increasing the bottom heat flux in the range of $q_{Tb}$=50–90 mW m$^{-2}$ facilitates the development of the thermohaline dome, increasing its size and thus the average temperature of the model (Fig. 6.a and c), as well as the
intensity of convection within the salt dome (Fig. 6.b). However, for higher heat fluxes ($q_{Tb}$>90 mW m$^{-2}$), the behaviour of the topothermohaline system changes. Intense thermal convection breaks down the thermohaline dome, increasing the relative area of the freshwater and brackish zone (Fig. 6.a). For the whole basin, this results in a faster flow, lower average temperature, lower salinity and younger waters (Fig. 6.b–e). Even though the cause is quite different, the behaviour of the control parameters is similar to the influence of the increasing water table amplitude. It is, therefore, important to emphasise that a higher bottom



heat flux can — paradoxically — also lead to lower temperatures, as vigorous thermal convection also drives heat itself out of the basin by breaking up the hot thermohaline dome beneath the discharge zone.

### 3.1.4 Effect of bottom solute concentration

Groundwater basins are often located above a low-permeability basement, whose salinity can vary strongly depending, for example, on the presence of evaporites, pore water salinity, etc. Figure 7 shows two snapshots of quasi-stationary numerical
solutions with prescribed solute concentration at the bottom boundary of the basin of $c_b$=10 g l$^{-1}$ (left panel) and $c_b$=200 g l$^{-1}$ (right panel). Other parameters agree with the values of the base model (Table 2). At low salinities, topography-driven forced convection and free thermal convection together control the flow regime. As a consequence, the negative haline buoyancy is negligible and no thermohaline dome is formed. Topography-driven regional flow is perturbed by warm plumes rising from the bottom of the basin and drifting towards the discharge zone (Animation S6 in Galsa et al., 2025). The majority of the basin
is saturated by cold, young freshwater, and the thermal plumes ascend to the surface unopposed, resulting in lower temperatures. The quasi-stationary plume of warm, brackish and aged water on the right side of the model domain is the result of the dynamic equilibrium of infiltrating cold water, rising warm water and a no-flow side boundary. In the model where high salinity is prescribed at the bottom of the basin (Fig. 7, right panel), the negative haline buoyancy is sufficiently large for the formation of a thermohaline dome. The high density of salt water effectively retains heat and discharge flux, resulting in
extremely high temperatures and water ages even in homogeneous basin, respectively. Spontaneous stratification develops inside the dome, with three layers characterized by separate thermohaline convection (Animation S7 in Galsa et al., 2025). In the multilayer thermohaline dome, the salt concentration and the age of the layers increase with depth.



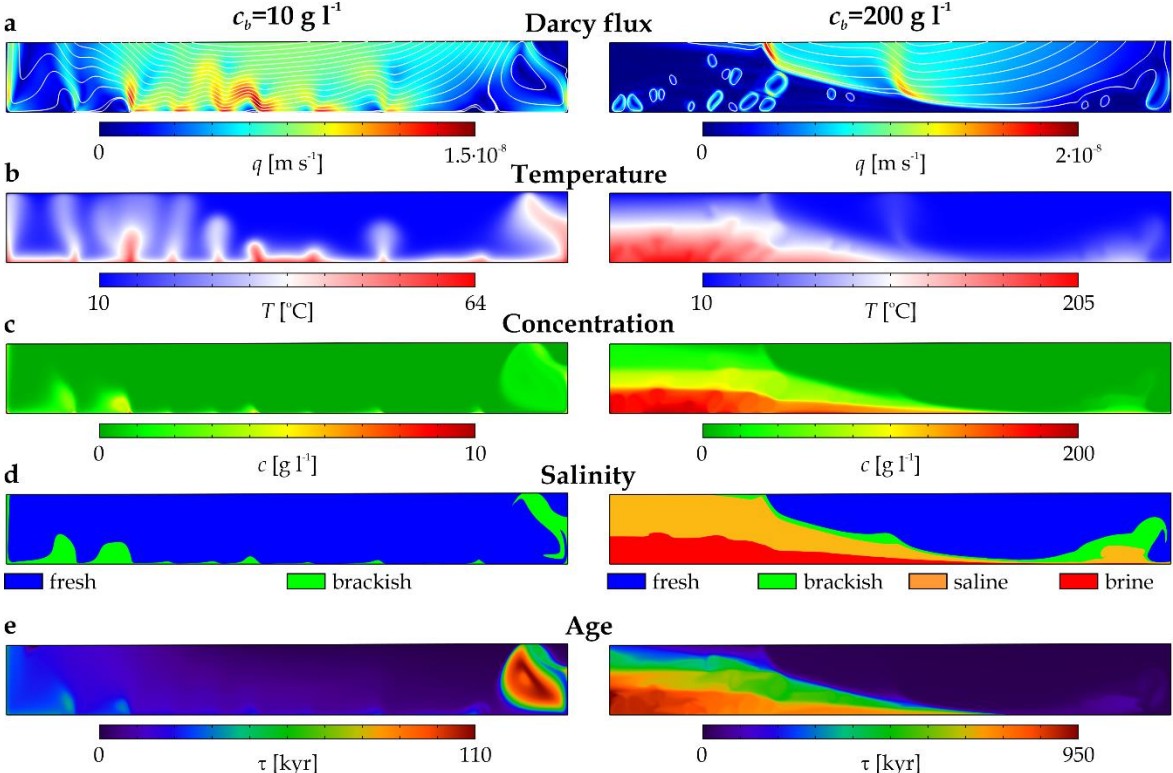

**Figure 7. Numerical solutions of (a) the Darcy flux, (b) the temperature, (c) the salt concentration, (d) the salinity zones and (e) the water age at the bottom salt concentration of $c_b$=10 g l$^{-1}$ (left panel) and $c_b$=200 g l$^{-1}$ (right panel). The quasi-stationary solutions are shown at 2 Myr.**

The control parameters indicate that as $c_b$ increases, the freshwater zone shrinks, although the reduction of the relative area ($A_{fre}$) stops at $c_b \geq 100$ g l$^{-1}$, and its extent does not fall below 40% even at extremely high salinity (Fig. 8.a). While $c_b$ increases by 2 orders of magnitude (from 2 to 200 g l$^{-1}$), the average salinity of the basin increases by more than 3 orders of magnitude from $c_{av}$=0.031 to 37.2 g l$^{-1}$ (Fig. 8.d). The explanation is that the high salinity slows down the flow (Fig. 8.b) and reduces the advective mass transport, which thus acts as a positive feedback. The mass transport is accompanied by a reduction in heat transport, which increases the average temperature of the basin by about 30 °C from $T_{av}$=21.6 to 53.4 °C, clearly caused by the appearance of saline and brine waters (Fig. 8.c). As the concentration of the lower boundary increases, the average water age of the basin increases by more than one order of magnitude (from $\tau_{av}$=14.4 to 218 kyr), also due to the increasing dominance of saline/brine and aged waters (Fig. 8.e). It is maintained that moving from the freshwater zone to the brine water zone slows down the flow, while the temperature, solute concentration (it is trivial) and age of the water gradually increase.





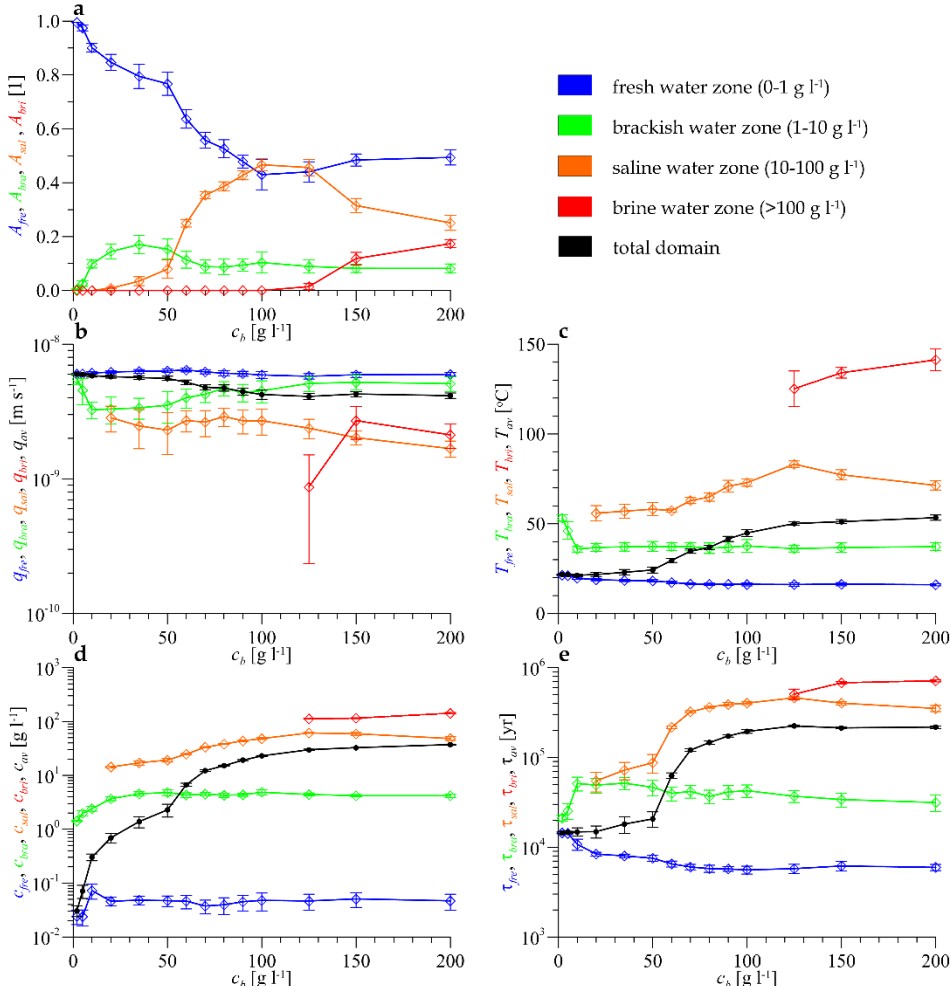

**Figure 8. (a) The relative area, (b) the Darcy flux, (c) the temperature, (d) the concentration and (e) the age of the salinity zones plotted against the bottom solute concentration. Each salinity zone is marked with a different colour. The values averaged for the whole model domain are shown in black. Standard deviations calculated for the quasi-stationary phase of the time series are marked by error bars.**

## 3.2 Topothermohaline convection in Buda Thermal Karst, Hungary

### 3.2.1 The Buda Thermal Karst system and its numerical model

The Buda Thermal Karst (BTK) system consists of two main parts: a predominantly unconfined and a confined carbonate system (Mádl-Szőnyi and Tóth, 2015). The two interconnected parts are bordered by the Danube River as a natural geographical margin. The uncovered karst west of the Danube includes the Buda Hills (559 asl) and the Pilis (756 asl), while the eastern side includes the Pest Plain and the Gödöllő Hills, where the topography varies between 95 and 350 asl (Fig. 9). The mean annual temperature of the Buda Thermal Karst system is 10 °C, while the annual precipitation is 500–600 mm yr$^{-1}$ (Mersich et al., 2003).



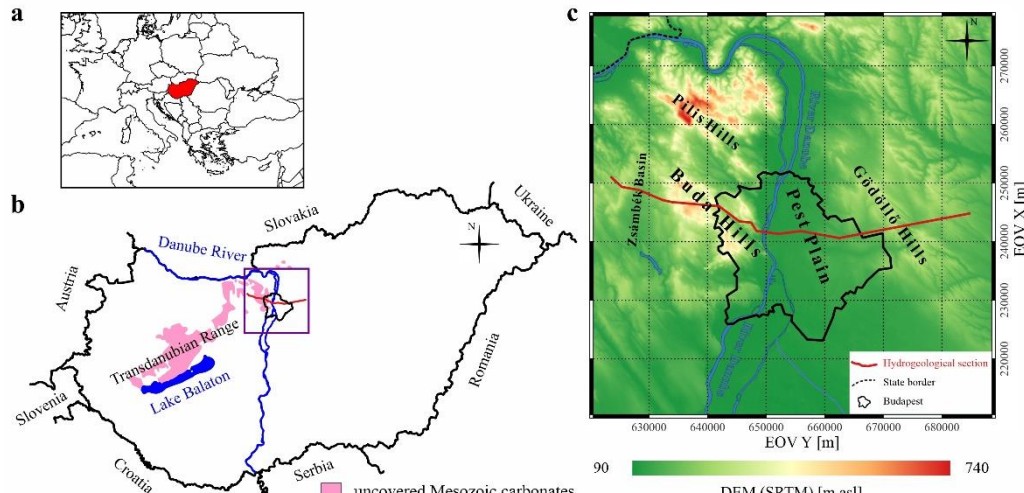

**Figure 9. (a) The location of Hungary in Europe, (b) the study area with uncovered Mesozoic carbonates and (c) the topography of the study area. The hydrogeological section crossing the Buda Thermal Karst system is marked by the red line (modified after Szijártó et al. (2021)).**

The western part of the BTK consists of 2–3 km thick, karstified dominantly Triassic limestone, dolomite and marl, and is, therefore, the main aquifer of the area (HS8–10 in Fig. 10). The system is downfaulted to the east of the Danube and becomes confined. The carbonate complex of the BTK is overlain by siliciclastic sediments and marls from the Eocene, as well as by pelagic clays from the Oligocene (HS4–7). The latter is the main aquitard unit in the area. From the Miocene, siliciclastic sediments began to be deposited again, alternating low and high permeability layers until the present surface (HS1–3) (Mádl-Szőnyi, 2019). Figure 9 shows the trajectory of the modelled hydrogeological section crossing Budapest, the capital of Hungary, and the wider geographical setting of the BTK system. Figure 10 presents the topography, the geology interpreted by Fodor (2011) and the hydrogeological conditions, including 11 hydrostratigraphic units (HS) and the simplified water table elevation (Mádl-Szőnyi et al., 2019; Szijártó et al., 2021) along the 2D section with a length of 63.8 km and average thickness of 5140 m. Table 3 summarizes the physical parameters of each unit, adapted from Szijártó et al. (2021), where the hydraulic conductivity of the layers, $K$ is given instead of the permeability, as only these were available from the literature and well tests. Minor modifications were introduced in the values of the hydraulic conductivity of HSs, since the model of Szijártó et al. (2021) applied isotropic hydraulic conductivity, whereas this model assumes a higher anisotropy ($\varepsilon = k_x/k_z = 100$) for siliciclastic units (e.g. Galsa, 1997) and a lower anisotropy ($\varepsilon = 10$) for carbonate units (e.g. Havril et al., 2016).







**Figure 10.** (a) The topography along the two-dimensional section, (b) the geological section (based on Fodor (2011)) and (c) the hydrogeological section, including the 11 HSs and the simplified water table adapted from Szijártó et al. (2021). Vertical exaggeration is 1.5.

**Table 3. Parameters of hydrostratigraphic units in the Buda Thermal Karst model.**

| Hydrostratigraphic units | $K$ [m s⁻¹] | $\varepsilon$ [1] | $\Phi$ [1] | $\rho_m$ [kg m⁻³] | $\lambda_m$ [W m⁻¹ K⁻¹] | $c_{pm}$ [J kg⁻¹ K⁻¹] |
|---|---|---|---|---|---|---|
| HS1 | $10^{-5}$ | 100 | 0.15 | 2200 | 1.8 | 840 |
| HS2 | $10^{-6}$ | 100 | 0.10 | 2200 | 1.8 | 840 |
| HS3 | $10^{-6}$ | 10 | 0.10 | 2600 | 2.0 | 900 |
| HS4 | $10^{-8}$ | 100 | 0.10 | 2500 | 1.8 | 900 |
| HS5 | $10^{-7}$ | 100 | 0.10 | 2500 | 1.8 | 900 |
| HS6 | $10^{-9}$ | 100 | 0.05 | 2000 | 1.5 | 2100 |
| HS7 | $10^{-6}$ | 100 | 0.15 | 2100 | 2.0 | 840 |
| HS8 | $10^{-5}$ | 10 | 0.15 | 2700 | 2.5 | 850 |





| HS9 | $10^{-5}$ | 10 | 0.15 | 2700 | 2.5 | 850 |
|---|---|---|---|---|---|---|
| HS10 | $10^{-5}$ | 10 | 0.10 | 2750 | 2.2 | 800 |
| HS11 | $10^{-7}$ | 10 | 0.10 | 2750 | 2.2 | 800 |

The boundary and initial conditions were chosen to be similar to those of the synthetic base model, which were certainly adapted to the specific hydrogeological situation. For the flow, no-flow boundary conditions were imposed at the

lower and vertical boundaries of the model, while the topography-driven groundwater flow was driven by the gradient of the water table determined on the basis of the available data (Fig. 10.c). The lateral groundwater connection between the BTK and the surrounding hydrogeological systems is not excluded, but no observations are available to quantify this process. The initial condition was a Darcy flux distribution corresponding to the reference water density. For temperature, the vertical boundaries were assumed to be insulating in the absence of significant transverse water movement, and a constant heat flux of 100 mW m$^{-2}$

was applied along the lower boundary due to the high heat flux characterizing the area (Lenkey et al., 2021). The surface temperature was set at 10 °C due to the geography of the study area. The initial condition was a conductive temperature distribution. For salt concentration, no flux at the side boundaries and constant concentration at the lower (35 g l$^{-1}$) and upper (0 g/l) boundaries were prescribed. As an initial condition, the concentration was uniform at 35 g l$^{-1}$ in all hydrostratigraphic units due to the fact that each formation was deposited in a marine environment (Mádl-Szőnyi et al., 2019). This is reflected

in the chosen lower boundary condition, while the upper boundary condition is the consequence of precipitation infiltration. For the water age, similar to the base model, there was no flux at all boundaries except on the surface, where the open boundary condition was applied (Eq. (7)), the water age at the beginning of the simulation was set to zero in the whole model domain.

Analogous to the synthetic model simulations, the numerical treatment of the BTK problem was obtained by solving the PDE system of Eqs. (1)–(5) using COMSOL Multiphysics 5.3a software. The maximum finite element size used was 100 m

inside the model domain and 50 m at the outer boundaries of the model, where boundary layer elements were also used. This resulted in a total of 160 775 elements. Unknown variables inside the finite elements were approximated in the same way as for the synthetic model. At the beginning of the time-dependent simulation, the time step was increased exponentially until 1 kyr to handle the initial transient phenomena more accurately, and then the maximum time step was fixed at $t$=20 yr. The simulation was continued to 1 Myr, because this time scale is comparable to the geological time scale of the area, and longer

calculations would have had to take geological processes into account (Havril et al., 2016). The calculation, therefore, took about 10 days CPU time and required approx. 8 GB of memory.

**3.2.2 Numerical model results of BTK system**

Figure 11 illustrates the evolution of the flow, temperature, solute content and water age of the Buda Thermal Karst system over 1 Myr. At 1 kyr after the start of the simulation, two types of flow regimes were developed (Fig. 11.a). (1) A topography-

dominated flow regime was formed in the unconfined part of the Buda Thermal Karst system ($x$=10–27 km in HS8–10) down to a depth of 3 km, and in the uppermost Neogene aquifers on the Pest side ($x$=30–63 km in HS1–2). (2) On the other hand,





thermohaline convection developed in the deeper, saline covered carbonates (e.g. $x$=47–62 km in HS8). After 1 kyr, most of the system is characterized by its initial state, with nearly conductive temperature distribution (Fig. 11.b) and high solute content (Fig. 11.c and d). The infiltrating young freshwater reaches only the upper few hundred meters of the near-surface

permeable units (Fig. 11.c–e). For better comparisons, the water age non-dimensionalised with simulation time is shown in Fig. 11.

After 10 kyr, precipitation infiltrating through unconfined carbonates floods the karstified reservoir (HS8–10) with young, cold freshwater (Fig. 11.g–j). Moreover, this water mass is already appearing in the confined reservoir, about 5 km east of the Danube River ($x$=32 km). At this stage, most of the confined carbonate reservoir (HS8) and the Oligocene clayey cover

(HS5–6) are still saturated with warm, saline water. In the confined karsts, thermohaline convection forms in the saline pore water at $x$=32–62 km (Figure 11.f).

100 kyr after the start of the simulation, the flow pattern is essentially unchanged (Fig. 11.k). The size of the topography-dominated flow region has increased, while the front of warm, saline and aged water associated with the confined carbonates has clearly retreated to the east (Fig. 11.l–o). The boundary of the thermal water is now 15 km east of the Danube

($x$=42 km), the saline water was replaced by brackish water up to the Szada Fault ($x$=47 km). There is a strong correlation between water age and salt concentration distribution (Fig. 11.m and o): the younger, less saline water is already present in the deepest covered carbonates (HS8: $x$=58 km). Thermohaline convection in the confined reservoirs effectively mixes young, cold freshwater infiltrating from the west with old, warm and saline water from the east, thus facilitating the propagation of the topography-driven groundwater flow zone.

After 1 Myr, the high-temperature geothermal reservoir ($T$>150 °C) retreats to the deeper (>2 km) eastern zone ($x$>42 km) (Fig. 11.q). Moreover, thermal water with medium temperature ($T$=50–150 °C) is found in the western part of the confined reservoir under the clayey cover layer (HS7–8: $x$=27–42 km), which reaches the surface at the Danube line at the confined-unconfined margin. The extent of the saline groundwater is somewhat larger, as they are found in the low permeability Oligocene covers in addition to the high-temperature geothermal reservoir (Fig. 11.s). The Oligocene clays (HS6)

contain the oldest waters of the section, with an age practically equal to the duration of the simulation (1 Myr), i.e. their origin is representative of the sedimentary environment (Fig. 11.t). The cold, young, low salinity waters infiltrating from the unconfined site reach as far as the Szada Fault ($x$=47 km), and even result in a noticeable decrease in concentration and rejuvenation of the groundwater in the karstified carbonates (HS8) up to the eastern boundary of the section (Fig. 11.r and t). In these domains, thermohaline convection is the prevailing component of groundwater flow over a long period of time (Fig.

11.p). In the Neogene sediments above the cover layer (HS1–2: $x$=32–63 km) and in the unconfined carbonates (HS8–10: $x$=10–27 km), topography-driven water flow continues to dominate, characterised by cold and young water with low solute content.







**Figure 11. Evolution of the topothermohaline convection in the Buda Thermal Karst system. Distributions of (a, f, k, p) the Darcy flux, (b, g, l, q) the temperature, (c, h, m, r) the solute concentration, (d, i, n, s), the salinity zones and (e, j, o, t) the water age at different simulation times of 1 kyr, 10 kyr, 100 kyr and 1 Myr. The direction of Darcy flux is shown with streamlines (white). The water age, τ is normalized by the simulation time, t.**

As concerns the Buda Thermal Karst system, it can be concluded based on the numerical model calculation that cold and young freshwater entering through unconfined karstic carbonates is effectively mixed with the warm, aged and saline water characteristic of the confined carbonate system below the Pest plain. The mixing zone propagates gradually but at a



slower rate eastwards, facilitated by thermohaline convection in the reservoir. Groundwater, mixed in temperature, chemical composition and age, moves westwards within the upper part of HS8 directly beneath the Oligocene cover, transporting the mixed water to the Danube line, the margin between the confined and unconfined carbonate system, where it reaches the surface (Animation S8 in Galsa et al., 2025).

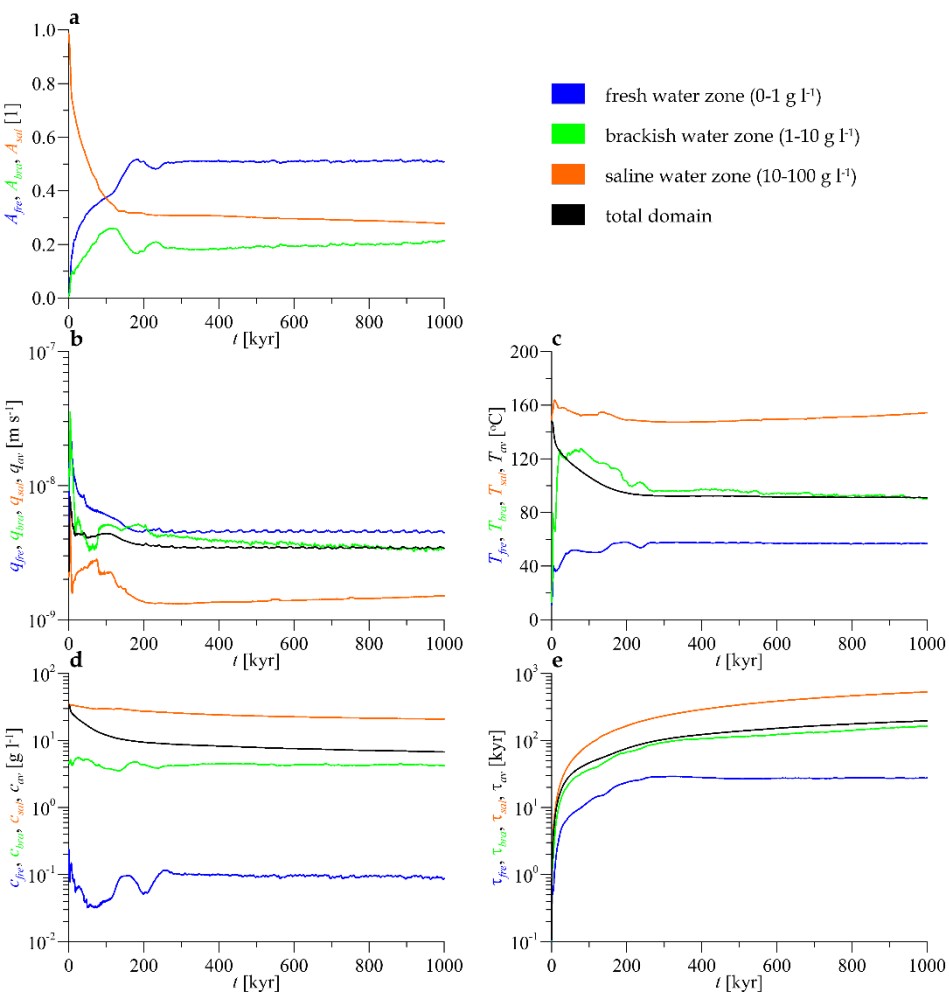

**Figure 12. The time series of (a) the relative area, (b) the Darcy flux, (c) the temperature, (d) the concentration and (e) the water age of different salinity zones. The values averaged for the whole model domain are shown in black.**


Most of the time series of the control parameters shows that the BTK model reached or approached its quasi-stationary solution after 200–300 kyr (Fig. 12). After this transient process, half of the system is saturated with freshwater ($A_{fre}$=51%), and the relative area of the brackish water domains is slowly increasing (currently $A_{bra}$=21%) at the expense of the saline water domains ($A_{sal}$=28%) (Fig. 12.a). The latter category includes, as illustrated in Fig. 11, the deep geothermal reservoir and the



Oligocene clay cover. The Darcy flux time series makes it clear that the solution of the problem is not stationary (Fig. 12.b). The periodicity of the flux intensity suggests that the processes in the freshwater domains in the first place and in the brackish water domains in the second place are weakly time-dependent. Their local Rayleigh numbers may slightly exceed the critical value, resulting in weak, periodic thermal convection in the deepest part of the unconfined carbonates and thermohaline convection in the confined carbonates (Animation S8 in Galsa et al., 2025). The saline, brackish and freshwater regions are

clearly distinct in terms of flow intensity, temperature and age (Fig. 12.b, c and e). The saline zones are characterised by slow flow ($q_{sal}=1.5\cdot10^{-9}$ m s$^{-1}$), high temperature ($T_{sal}=154$ °C) and high water age ($\tau_{sal}=530$ kyr). These regions saturated with saline and brackish water have not completely reached the quasi-stationary state even after 1 Myr. In contrast, freshwaters, directly connected to precipitation, have a higher and time-varying Darcy flux ($q_{fre}\approx4.4\cdot10^{-9}$ m s$^{-1}$), but lower temperature ($T_{fre}=57$ °C) and age ($\tau_{fre}=27.5$ kyr). Based on the control parameters, the most marked differences between the saline and freshwater

domains are in solute concentration (factor of 240) and age (factor of 19.3). These parameters seem to be the most sensitive for the separation of the different zones (Fig. 12.d and e).

## 4 Discussion

We performed a two-dimensional synthetic simulation set to assess the interaction of topography-driven forced convection and free thermal and haline convection on the groundwater flow. In the studied base model (Tables 1 and 2), two flow domains

are separated, the recharge and transition zones are dominated by topography-driven regional water flow, and the domain is characterised by young, cold freshwater. The emerging warm and brackish upwellings are driven by regional flow towards the discharge zone. However, a thermohaline dome develops beneath the discharge zone, in which free thermohaline convection is the dominant flow regime with warm, saline and aged waters (Fig. 2).

The reduction of the hydraulic gradient strengthens the thermohaline flow regime, which, in fact, floods the entire

basin at low water amplitude ($H\leq20$ m). In a narrow interval ($H=20–100$ m), the role of the thermohaline flow regime is strongly reduced, and in the models of $H=200–500$ m, the influence of regional flow is clearly dominant over most of the basin ($A_{fre}>95\%$). This sweeps warm, saline, aged waters out of the model domain, and also results in a near steady-state solution (Fig. 3). For these simulations, ignoring free thermohaline convection would only lead to minor inaccuracies in the solutions.

The variation of the bottom heat flux affects the behaviour of the coupled flow system directly through the

temperature. It is not evident that in the model of $q_{Tb}=90$ mW m$^{-2}$, the extension of the thermohaline dome is at its maximum ($A_{sal}=35\%$), and the average temperature is at its highest ($T_{av}=35$ °C). A higher bottom heat flux induces such intense thermal convection that it leads to a weakening/disappearance of the thermohaline dome and reduces the average temperature of the basin paradoxically (Fig. 5 and 6).

At low bottom salt concentrations, the flow in the basin is controlled by the topography of the water table, which is

perturbed by buoyant thermal upwellings (Fig. 7). In models with $c_b\leq20$ g l$^{-1}$, a large part of the basin is saturated by freshwater ($A_{fre}>80\%$), resulting in low average temperature ($T_{av}<22$ °C) and young water ($\tau_{av}=15$ kyr). Increasing $c_b$ moderates the flow



and forms a dome with multilayer thermohaline convection beneath the discharge zone. This is accompanied by temperature and water age increases of more than 30 °C and one order of magnitude, respectively.

510          Simple (two-dimensional, homogeneous and isotropic) synthetic model calculations draw attention to the fact that the final state of the interconnected processes in nature is, in most cases, not predictable. While the effect of changing some parameters can be qualitatively inferred (e.g. increasing the amplitude of the water table enhances the dominance of topography-driven groundwater flow), quantification is only possible on the basis of numerical, possibly analytical, calculations. In addition, there are complex relationships in the coupled problem whose effect is not even qualitatively evident (increasing heat flux decreases average temperature). In this context, it can be concluded that the numerical simulations should

be carried out by building the most general model possible, of course, after compromising with the available computational resources. This will ensure that the subjectivity of the numerical modeller has the least possible influence on the final results of the calculations. However, depending on the hydrogeological and geothermal conditions of the area, certain phenomena and physical processes can be ignored, but their negligibility must always be confirmed by a series of comprehensive preliminary simulations.

520          The operation of the topothermohaline convection model was demonstrated in a real hydrogeological situation in the Buda Thermal Karst area of Hungary. The numerical solution suggests that the western, uncovered karst is filled up by cold, young freshwater infiltrating from precipitation in as short as 10 kyr (Fig. 11.f–j). Here, the topography-driven forced convection is the dominant driving force, apart from warm upwellings evolving in the deepest parts of the system. However, groundwater with higher temperature and salinity in the eastern confined karst may persist over geological time scales, at least

in the more confined eastern zones of the reservoir (Fig. 11. p–t). In these confined but highly permeable regions, free thermohaline convection develops, actively transporting warm, solute-rich and aged waters towards the main outflow zone, the Danube River. Thus, waters of different temperature, chemical composition and age from the western unconfined karst and the eastern confined karst are effectively mixing under the western part of the Pest Plain, contributing to the karstification process (Erőss et al., 2008; 2012; Erőss, 2010; Licour, 2014).

530          This picture is strongly analogues to the flow-temperature-salinity system in the Tiberias Basin revealed by the numerical simulations of Magri et al. (2015), although that area is characterised by significantly lower heat flux (60 mW m$^{-2}$) but higher TDS content (up to 300 g l$^{-1}$) and different geological built-up. The near-surface Quaternary and Tertiary complexes are dominated by topography-driven regional groundwater flow with low temperature and salinity, while thermohaline convection is formed in the deeper and more confined Mesozoic carbonates of the Golan Heights. The connection of the latter

to the surface is through high permeability faults, which explains the higher temperatures and TDS contents in springs and wells around Lake Tiberias.

         In the case of BTK, lukewarm springs in uncovered karst areas are mainly characterised by intermediate temperature (20–28 °C), low chloride content (10–40 mg l$^{-1}$) and high yield (>1800 l min$^{-1}$). However, as the margin of the uncovered-covered karst is approached, thermal springs (38–58 °C, >80 mg l$^{-1}$, <600 l min$^{-1}$) appear. The analysis of the chemical

composition of the spring waters clearly indicates a discharge of two different types of water (Kovács and Erőss, 2017;Mádlné



Szőnyi et al., 2018; Mádl-Szőnyi, 2019). New geothermal projects in the BTK area generally target fresh or brackish thermal waters with a moderate temperature (50–80 °C) in the western part of the shallower, confined karst (HS7 and 8: $x$=27–42 km). This is fully consistent with the numerical model of westward groundwater flow under the Oligocene clayey cover, which transports the groundwater having mixed chemical composition, temperature and age from the eastern reservoir (Fig. 11.p).

Nevertheless, it highlights the significant geothermal potential of the upper Triassic Dachstein Limestone (HS8) and Eocene (HS7, e.g. Szépvölgyi Limestone) reservoirs in the eastern part of the BTK (Fig. 10.b), which is of higher temperature but greater depth, according to our numerical simulations.

Szijártó et al (2024) studied the water age of the BTK system in a geologically simplified two- and three-dimensional numerical model. He found that the calculated water age in the confined reservoir, especially in its eastern and deeper regions,

increases by orders of magnitude ($\tau$>100 kyr, in some cases >1 Myr) compared to the shallower regions of the unconfined area ($\tau$=0–10 kyr), in full agreement with the results presented here. This is partly supported by the highly sporadic records of [14]C ages of water samples from springs, caves and thermal wells. The ages of spring waters in uncovered areas show approximate values of 5 kyr, while the ages of waters from the depth range of 1000–1600 m suggest values of 15–25 kyr (Deák, 1979; Fórizs et al., 2019). Our numerical model results are consistent with the water ages measured in the shallow and uncovered

areas but produce somewhat higher water ages in the deeper zones (~50 kyr). Overall, Balderer et al (2014), based on [36]Cl/Cl ratio data, can reflect equilibrium conditions and long (~million years) residence time. They propose the possibility of mixing with "very old" groundwater components.

Of course, like all models, the present simulations apply approximations too. Although the dependence of the water density on temperature, pressure and salinity in the buoyancy was taken into account, the effect of these parameters on the

viscosity of the water was neglected. Based on our preliminary calculations, the effect of temperature is the dominant factor that promotes transport processes in the deeper, warmer regions, thus increasing the intensity of thermohaline convection in confined reservoirs. Meanwhile, in the unconfined zones, low temperature slightly moderates the flow intensity due to enhanced viscosity. The two-dimensional approach obviously reduces the degree of freedom of the system, although the 3D treatment of the phenomenon is not yet feasible with the available computing capabilities (1–2 weeks CPU time for one

simulation in 2D). Nevertheless, two-dimensional simulation is an effective way to facilitate the preliminary investigation of real hydrogeological problems and to draw attention to the complexity of the phenomenon and the interaction of the different forces driving groundwater flow.

**5 Summary and conclusions**

Two-dimensional numerical model calculations were performed to reveal the combined effects of topography-driven forced

convection and temperature- and salinity-driven free thermal and haline convection on the basin-scale groundwater flow. In the first step, we analysed the influence of the amplitude of the groundwater table, the bottom heat flux and salt concentration on the flow pattern, temperature, solute content and water age distribution in a synthetic homogeneous model domain. All of





the factors investigated significantly affect the control parameters, and the simulations lead to a time-dependent quasi-stationary solution with a dynamic equilibrium between topography-driven forced convection, positive thermal and negative

haline buoyancy. We conclude that

- in the base model ($H$=50 m, $q_{Tb}$=90 mW m$^{-2}$, $c_b$=70 g l$^{-1}$), the recharge zone and the central regions of the basin are dominated by topography-driven regional groundwater flow, with low average temperature, salinity and water age. While below the discharge zone, a dome with high temperature, salinity and water age develops, in which vigorous thermohaline convection is formed.

- Decreasing the amplitude of the water table reduces the hydraulic driving force and, thus, facilitates the extension of the thermohaline dome, which can eventually flood most of the basin. In contrast, increasing the water table amplitude intensifies the topography-driven regional water flow, which effectively sweeps heat, solute content and aged water out of the system. This shifts the time-dependent solution towards a stationary final state.

- The decrease in the bottom heat flux reduces the thermal buoyancy, thus the extent of the thermohaline dome, the

flow intensity and the average temperature. Increasing the heat flux, on the other hand, strengthens the thermal buoyancy and, thus the intensity of the water flow, which also leads to a decrease in the average temperature and the rejuvenation of the groundwater.

- The decrease in salt concentration at the bottom boundary weakens the role of negative haline buoyancy, resulting in the dominance of topography-driven flow, perturbed by warm upwellings drifting towards the discharge zone.

Increasing salt concentration restrains regional flow and promotes the development of a multilayered thermohaline dome, which significantly enhances both the average temperature and the water age in the basin.

The physical understanding of the combinations of driving forces sheds light on the complex processes that control flow in groundwater basins. It highlights (i) the large variations in temperature, salinity and water age as a result of topothermohaline convection, even in homogeneous basins; (ii) the importance of understanding the fluid evolution history of

sedimentary basins based on the time-dependence of these processes, and (iii) the often oversimplified flow interpretations derived from sporadic water chemistry or isotopic data.

The phenomenon of topothermohaline convection was demonstrated along a real two-dimensional section crossing the Buda Thermal Karst, Hungary. The section is divided into two areas with different characters by the Danube River. West of the Danube, karstified carbonates are mostly unconfined, while east of the Danube, they are covered by the Oligocene clayey

and Neogene sediments. Consequently, numerical model results show that

- in the western unconfined area, as little as 10 kyr is sufficient to saturate a large part of the karstified carbonates with infiltrating cold, young, fresh precipitation. The flow is primarily controlled by the topography of the groundwater table, only in the deepest parts of the basin do warm upwellings develop and drift towards the Danube as the main discharge zone.

- In contrast, in the deep and confined karst reservoir of the eastern side, high temperatures, salinity and water age can persist over geological timescales (>1 Myr), making the area a prime target for future geothermal exploration.



Thermohaline convection forms in the covered carbonates, transporting hot, saline and aged waters toward the Danube.

- Between the two areas, but already in the confined part, a mixing zone is formed. The mixed temperature and chemistry of the groundwater contribute to the evolution of karstification in the confined reservoir and explain the difference between the surface lukewarm and warm springs as well as the temperature distribution in the thermal wells.

The gradual increase in geological and geophysical observations from the BTK site will allow the model to be fine-tuned, and, thus the parameters of the covered geothermal reservoir on the eastern side to be more accurately determined.

## Data availability

No data sets were used in this article

## Author contributions

AG: conceptualization, investigation, methodology, investigation, resources, validation, visualization, writing – original draft. MSz: data curation, funding acquisition, resources, supervision, visualization. ÁT: data curation, supervision. JM-Sz: data curation, conceptualization, funding acquisition, supervision.

## Competing interest

The authors declare that they have no conflict of interest.

## Acknowledgements

The research was funded by the National Multidisciplinary Laboratory for Climate Change, RRF-2.3.1-21-2022-00014 project and by the National Research, Development and Innovation Office in the framework of project No. PD 142660.



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
