# Peer review of "Topothermohaline convection – from synthetic simulations to reveal processes in a thick geothermal system"

_EGUsphere, 2025_

## Author Comment (AC1)

Dear Xiaolang Zhang,

First of all, thank you very much for your valuable comments. In this phase, we will answer the questions you raised, and the revised manuscript will be attached later, after we received all reviews.

**General comment:**
Some papers dealing with the combined effects of water table topography, thermal and haline buoyancy were mentioned in the Introduction (line 102–109: Hoyos et al., 2012; Gupta et al., 2015; Magri et al. 2015). The only one of these was the paper by Magri et al. (2015), whose results could be compared in a meaningful way, and we did so in the Discussion (line 530–536). Agreeing with your suggestion, other connecting papers (e.g. Zech et al., 2016; Kaiser et al., 2013a) will be discussed as well as potential areas will be presented, where considering the effect of topothermohaline convection, might be warranted in the future. This will be done in the revised manuscript.

**In the Abstract, "water table topography" should be revised to "water table undulations."**
We accept that 'water table undulation' is a slightly more accurate term to describe the phenomenon than 'water table topography'. However, topography-driven groundwater flow is the common expression (Mádl-Szőnyi et al., 2023), and topothermohaline convection includes this part. Thus, we would like to keep the original term, especially in the first sentence of the Abstract. In the first part of Introduction, a brief correspondence is made between water table topography and undulation.

**Line 35: Should "horizontal variations in the water table" be "vertical variations"?**
In classical Tóthian models (Tóth 1962;1963), where the upper boundary of the model was horizontal ($h_{wt}(x)$), clearly the horizontal variation of the water table ($\partial h_{wt}(x)/\partial x$) results in groundwater flow. However, in recent numerical models (like those presented in this manuscript), where the upper flow boundary of the model geometry follows the water table, it has (a minor) vertical variation as well resulting in a slight difference between the numerical solutions. Since this part of Introduction reflects the first mathematical handling of the problem solved by prof. József Tóth, we would like to keep the original version.

**Line 38: Revise "thereof."**
To clarify, a minor modification was made in the sentence: 'can be considered as its subdued replica'.

**Line 144: How do you account for the relationship between salinity and density?**
Water density depends on the temperature (6[th] order polynomial), the pressure (2[nd] order polynomial) and the salt concentration (1[st] order polynomial) (line 160–162). A linear relation between the water density and the concentration of

$$\rho_w(c) = \rho_{ref}(1 + \beta c)$$

was applied after the recommendation of Kohfahl et al. (2015), where $\beta = 7.1\cdot10^{-4}$ l/g. This term was built in the whole formula of $\rho_w(T,p,c)$ which will be quantified in a new Supplementary material.

**Line 173: How do you ensure that the model discretization (time step and spatial mesh size) meets the requirements for the Peclet number and Courant number?**

For the simulations, we used COMSOL Multiphysics 5.3a, which is a finite element software package to solve coupled partial differential equation systems (Zimmerman, 2006). For discretization of time stepping in COMSOL, the Backward Differentiation Formula (BDF) was chosen, which is suggested for diffusion and convection problems due to its stability and robustness. To control BDF,

- the maximum time step was fixed at 100 yr for synthetic model calculation and 20 yr for simulating topothermohaline convection along a 2D section in the Buda Thermal Karst system having more complex geometry.
- A backward Euler scheme with an order of 1 or 2 was chosen to generate linear equation system, which was solved by a direct solver (MUMPS). If the global error (including both time stepping error and algebraic solution error) obtained from BDF exceeded a threshold (controlled by both relative and absolute tolerance), the time step was refused and
    i.   the size of the time step was decreased (typical), or/and
    ii.  the order of time discretization was increased
  to reduce the global error (summing up local errors) and to ensure the accuracy and the robustness of the numerical solution.
- In addition, both the non-dimensional Courant and Péclet numbers were calculated for each finite element and time step to check manually the local distribution of spatial and temporal quality of numerical solution.
- For example, the cell Péclet number, $Pe$ was calculated for the mass transport (Darcy flux*cell size/molecular diffusion), which exceeded the value calculated for heat transport by orders of magnitude. In this way, $Pe$ was much larger than 2 as a classical limit to characterize the accuracy of spatial discretization. However, this strict criterion is rarely used in practice, as it would require a very small element size. Instead, we applied consistent stabilization (crosswind and streamline diffusion), as a trade-off between sufficient accuracy and adequate element size (e.g. Diersch and Kolditz, 2002).
- Finally, a 1D stationary analytical model was built up to check the spatial accuracy of age transport simulation (Szijártó et al. 2025). Comparison between analytical and numerical solution of water age proved that the average and maximum relative deviations are below $10^{-3}$% and $10^{-1}$%, respectively, even for $Pe>>2$.

**Line 184: The boundary condition for a constant salinity concentration seems unreasonable. While the heat source can originate from deep geologic units, where does the constant salt flux come from?**

In advance, all boundary conditions are artificial and only an approximation of reality. In Hungary, it is typical that the salt concentration of water increases with depth. It is a consequence of (1) the sedimentation that took place predominantly in a marine, saline water environment (line 404) and (2) the precipitation providing recharge that mainly reduces the salinity of the shallower layers. Consequently, a basement beneath the aquifer modelled has a higher salinity due to its lower permeability. It is worth noting that the model itself clearly shows that the low-permeability clayey Oligocene layer is able to retain its original salinity (HS6 in Fig. 11), despite (1) being at relatively shallow depths and (2) the presence of high-permeability aquifers below and above it. In addition, there are a number of hydrogeological situations where the aquifer is located above evaporites, halite formations (e.g. Kaiser et al, 2011; 2013b; Gupta et al. 2015; Magri et al., 2015; Zech et al., 2016), and thus salt source from a lower aquitard is almost constantly ensured. Nevertheless, the constant salt concentration prescribed along the bottom boundary was varied in a very wide range from 2 to 200 g/l,

including many hydrogeological situations. Furthermore, a new model will be presented in the Supplementary material, in which the evolution of the BTK system will be investigated using (1) different lower boundary condition for the salt concentration and (2) temperature-, pressure- and concentration-dependent water viscosity.

Finally, we emphasize that the salinity concentration along the bottom boundary was constant during the simulations and not the salt flux.

**Figure 10: The caption font is too small to read.**

Thank you, the indexes are clearly readable in the original file format, but pdf generation deteriorated its quality. We will take care of it and now we attach the original figure.

References

de Hoyos, A., Viennot, P., Ledoux, E., Matray, J.-M., Rocher, M., and Certes, C.: Influence of thermohaline effects on groundwater modelling – Application to the Paris sedimentary Basin, *Journal of Hydrology*, 464–465, 12–26, https://doi.org/10.1016/j.jhydrol.2012.06.014, 2012.

Diersch, H.-J.G., and Kolditz, O.: Variable-density flow and transport in porous media: approaches and challenges, *Advances in Water Resources*, 25, 899–944, https://doi.org/10.1016/S0309-1708(02)00063-5, 2002.

Gupta, I., Wilson, A.M., and Rostron, B.J.: Groundwater age, brine migration, and large-scale solute transport in the Alberta Basin, Canada, *Geofluids*, 15, 608–620, https://doi.org/10.1111/gfl.12131, 2015.

Kaiser, B.O., Cacace, M., and Scheck-Wenderoth, M.: 3D coupled fluid and heat transport simulations of the Northeast German Basin and their sensitivity to the spatial discretization: different sensitivities for different mechanisms of heat transport, *Environmental Earth Sciences*, 70, 3643–3659, https://doi.org/10.1007/s12665-013-2249-7, 2013b.

Kaiser, B.O., Cacace, M., and Scheck-Wenderoth, M.: Quaternary channels within the Northeast German Basin and their relevance on double diffusive convective transport processes: Constraints from 3-D thermohaline numerical simulations, *Geochemistry, Geophysics, Geosystems*, 14 (8), 3156–3175, https://agupubs.onlinelibrary.wiley.com/doi/full/10.1002/ggge.20192 , 2013a.

Kaiser, B.O., Cacace, M., Scheck-Wenderoth, M., and Lewerenz, B.: Characterization of main heat transport processes in the Northeast German Basin: Constraints from 3-D numerical models, *Geochemistry, Geophysics, Geosystems*, 12, paper: Q07011, https://doi.org/10.1029/2011GC003535, 2011.

Kohfahl, C., Post, V.E.A., Hamann, E., Prommer, H., and Simmons, C.T.: Validity and slopes of the linear equation of state for natural brines in salt lake systems, *Journal of Hydrology*, 523, 190–195, http://dx.doi.org/10.1016/j.jhydrol.2015.01.054, 2015.

Mádl-Szőnyi, J., Batelaan, O., Molson, J., Verweij, H., Jiang, X.-W., Carrillo-Rivera, J.J., and Tóth, Á.: Regional groundwater flow and the future of hydrogeology: evolving concepts and communication, *Hydrogeology Journal*, 31, 23–26, https://link.springer.com/article/10.1007/s10040-022-02577-3, 2023.

Magri, F., Inbar, N., Siebert, C., Rosenthal, E., Guttman, J., and Möller, P.: Transient simulations of large-scale hydrogeological processes causing temperature and salinity anomalies in the Tiberias Basin, *Journal of Hydrology*, 520, 342–355, https://doi.org/10.1016/j.jhydrol.2014.11.055, 2015.

Szijártó, M., Galsa, A., Czauner, B., Erőss, A., Tóth, Á., and Mádl-Szőnyi, J.: Numerical investigation of groundwater ageing and thermal processes in confined-unconfined full basins with asymmetric flow pattern, *Hydrogeology Journal*, accepted.

Tóth, J.: A theoretical analysis of groundwater flow in small drainage basin, *Journal of Geophysical Research*, 68 (16), 4795–4812, https://doi.org/10.1029/JZ068i016p04795, 1963.

Tóth, J.: A theory of groundwater motion in small drainage basins in central Alberta, Canada, *Journal of Geophysical Research*, 67 (11), 4375–4387, https://doi.org/10.1029/JZ067i011p04375, 1962.

Zech, A., Zehner, B., Kolditz, O., and Attinger, S.: Impact of heterogeneous permeability distribution on the groundwater flow systems of a small sedimentary basin, *Journal of Hydrology*, 532, 90–101, http://dx.doi.org/10.1016/j.jhydrol.2015.11.030, 2016.

Zimmerman, W.B.J.: Multiphysics Modeling with Finite Element Methods, World Scientific Publishing Company, Singapore, p. 432. ISBN: 9812568433, https://doi.org/10.1142/6141, 2006.

---

## Author Comment (AC3)

Dear Yipeng Zhang,

We thank you for your review of the manuscript and for your comments, corrections and suggested modifications. These comments have been carefully considered and are responded to below:

**Major concerns:**
**First, the bottom boundary condition for salt is assigned to be constant value, which means that there is unlimited salt comes from underlying aquifer/aquitard. The reason to assign a constant salt boundary condition should be explained, and the potential influences of using such boundary condition on the result should be at least mentioned.**

Reviewer 1 also addressed this problem, to which we gave a similar response. In basin areas where sedimentation has taken place in a marine environment, such as the Pannonian Basin, it is a common hydrogeological situation that salinity generally increases with depth. The main reason for this is that the near-surface geological environment is in active contact with precipitation through recharge zones, while the deeper, more confined areas are much less so. Hence, the salinity of the aquitards below the basin exceeds that of the aquifers above, providing a continuous salinity supply to the basin. The phenomenon develops spontaneously in the BTK system itself, as the higher salt concentration in the low-permeability Oligocene cover persists over geological time, although it is interbedded with high-permeability aquifers from above and below. In addition, there are several hydrogeological situations where the bedrock of the basin is formed by evaporites, halite formations (e.g. Kaiser et al, 2011; 2013; Gupta et al. 2015; Magri et al., 2015; Zech et al., 2016), increasing the salinity of the basin through dissolution. Furthermore, in the synthetic parameter analysis, we varied the salt concentration of the lower boundary over a very wide range, $c_b$=2–200 g l$^{-1}$, to ensure that almost all hydrogeological situations were included.

Unfortunately, for the BTK system, there is no geological/geophysical/geochemical information available on the basement, and thus its exact salinity is not known. Therefore, a boundary condition corresponding to the initial condition (sediment deposition in a marine environment) was imposed in the model, $c_b$=35 g l$^{-1}$. However, as a consequence of the concern raised, a new simulation was also performed, in which an impermeable layer is located beneath the HS11 Lower Triassic-Paleozoic Aquitard in the BTK model, and a constant boundary condition of $c_b$=35 g l$^{-1}$ is prescribed at its lower boundary, allowing only diffuse bottom salt transport. As a result, the salinity of the confined geothermal reservoir decreased, thermohaline convection intensified, and the groundwater mean age decreased. At the same time, the nature of the complex flow regime that developed in the BTK system did not change: topography-driven groundwater flow in the unconfined karst area, thermohaline convection in the confined karst, eastern reservoir with high geothermal potential, etc. The newly performed simulation and its interpretation can be found in the new Supplementary material attached to the manuscript.

**Second, how is the application model in the BTK system verified to be representative for the pattern in the real system. Also, some discussion on the BTK system model should be added to show their implications in other regions globally.**
Agreeing with the suggestion, the Discussion was substantially expanded. In the discussion part, we have already compared the numerical solution of the BTK with (1) the temperature and salinity of observed springs, (2) the temperature and salinity of geothermal projects at shallower depths, and (3) available water age data. Furthermore, we have pointed out the strong analogy between the flow-temperature-salinity regimes of the BTK and the Tiberias Basin to emphasize that the Buda Thermal Karst system is not a unique hydrogeological formation. In the revised

version of the manuscript, both sections are expanded and completed to enhance the linkage of our research to other hydrogeological systems around the world. Additional salinity observations and temperature-elevation profiles are compared to our numerical results, and two other studies (Zech et al., 2016; Kaiser et al. 2013) are presented to illustrate the connection between processes of thermohaline convection occurring in BTK and other regions.

**Specific comments:**
**Line 11 It is weird to use water table topography, find a better word. Please also revise them accordingly in the later part of the manuscript.**
Reviewer 1 suggested the use of 'water table undulation', and a short explanation was inserted in the Introduction for clarification. However, the usual scientific term to describe this phenomenon is 'topography-driven groundwater flow' (e.g. Mádl-Szőnyi et al., 2023), which has been modified to emphasize that it is not the topography of the surface but the topography of the water table that causes forced convection. In addition, in the expression of 'topothermohaline convection', the first term thus directly refers to the effect of the water table topography. For these reasons, we would like to keep the original expression 'water table topography'.

**Line 13 Replace "combined" with "coupled".**
Replaced, thank you for clarification.

**Line 37 What is Robinson and Love (2013) improved?**
Robinson and Love (2013) accomplished an analytical stagnation point analysis and investigated the flow pattern asymmetry in a hierarchically nested groundwater system based on the Tóthian unit basin (Tóth 1963). We have inserted a short explanation into the text and added the missing reference.

**Line 70 What is "increase heat transport" mean?**
We have modified the expression in the revised manuscript: 'intensifies heat transport'.

**Line 132 Replace "balance" with "equilibrium".**
Done in the revised manuscript.

**Line 157 & Table1 Why are the longitudinal and transverse dispersivity set to 0 m in the synthetic models, and what is the potential influence of setting dispersivity to 0 m.**
In the synthetic models, we aimed to focus attention on the phenomenon of topothermohaline convection, so we compiled the simplest possible model framework, choosing to eliminate the effect of mechanical dispersion. At the same time, we have already taken into account the heterogeneity in nature in the real BTK system ($\alpha_L$=100 m, $\alpha_T$=10 m). In response to the question raised, we incorporated longitudinal and transverse dispersivity ($\alpha_L$=100 m, $\alpha_T$=10 m) into the synthetic base model (Table 2), and the model results are presented and interpreted in the Supplementary material attached to the manuscript, and the effect of dispersion is briefly summarized in the Discussion section.
In short, mechanical dispersion has strengthened the thermohaline dome that forms beneath the discharge zone, increasing its relative size from $A_{sal}$=35% to 50%. This resulted in an increase in the average salt concentration, temperature and water age of the basin, while the average Darcy flux decreased. The change is due to two factors: (1) transverse dispersivity increases the salt flux entering the bottom of the basin, and (2) longitudinal dispersivity effectively mixes the waters in the basin. By the way, the effect of mechanical dispersivity on topohaline

(topography-driven forced & haline buoyancy-driven free) convection was analysed in detail by Galsa et al. (2022).

**Line 164 Replace "supposed"**
Replaced to 'proposed'

**Line 170 The quantitative relationship between groundwater density, salinity, and temperature should be incorporated.**
The quantitative relation of the water density and the molecular diffusion depending on the temperature, pressure and salt concentration is presented in the new Supplementary material.

**Line 210 Please include references to support the selected values used in the sensitivity analysis.**
Yes, references have been inserted into the text to support the parameter ranges selected. Note that the parameter ranges used in the sensitivity analysis are so wide that they cover almost all hydrogeological situations.

**Line 391 Have the hydrogeological parameters of the fault been characterized separately? A brief discussion on the potential effect of fault in upwelling old, warm and saline groundwater in the discharge area should be added.**
In this study, we have not investigated the role of the conduit/barrier faults. The faults shown in Figures 10 and 11 only separate the individual geometric units without any specific physical parameters. However, Szijártó et al. (2021) studied the role of conduit faults in the 'topothermal model' of the BTK system and found that they can influence the temperature distribution and the flow field mainly locally. If conduit faults were present in the environment of a high-temperature reservoir, they could, in theory, cause local anomalies both in temperature and salinity, but such surface manifestations do not exist. Only at the margin of the confined and unconfined karst, along the Danube River, the occurrence of springs with different temperatures, chemical composition and yields can be seen even near each other, as already mentioned in the Discussion of the manuscript. To clarify the potential role of faults, a separate paragraph has been dedicated to this point in the Discussion.

**Figures 6, 8 &12 Some words in the figures are difficult to read, possibly due to the use of the color light green. Consider adjusting the color or increasing the font size.**
If the manuscript is accepted, we will definitely modify this colouring on the graphs, just as we will probably have to change some of the colour scales.

Furthermore, minor stylistic and grammatical errors have been corrected in the revised manuscript, which — at the request of the editor — has not been uploaded at this stage of the review.

**References**

Gupta, I., Wilson, A.M., and Rostron, B.J.: Groundwater age, brine migration, and large-scale solute transport in the Alberta Basin, Canada, *Geofluids*, 15, 608–620, https://doi.org/10.1111/gfl.12131, 2015.

Kaiser, B.O., Cacace, M., and Scheck-Wenderoth, M.: Quaternary channels within the Northeast German Basin and their relevance on double diffusive convective transport processes: Constraints from 3-D thermohaline numerical simulations, *Geochemistry, Geophysics, Geosystems*, 14 (8), 3156–3175, https://agupubs.onlinelibrary.wiley.com/doi/full/10.1002/ggge.20192, 2013.

Kaiser, B.O., Cacace, M., Scheck-Wenderoth, M., and Lewerenz, B.: Characterization of main heat transport processes in the Northeast German Basin: Constraints from 3-D numerical models, *Geochemistry, Geophysics, Geosystems*, 12, paper: Q07011, https://doi.org/10.1029/2011GC003535, 2011.

Magri, F., Inbar, N., Siebert, C., Rosenthal, E., Guttman, J., and Möller, P.: Transient simulations of large-scale hydrogeological processes causing temperature and salinity anomalies in the Tiberias Basin, *Journal of Hydrology*, 520, 342–355, https://doi.org/10.1016/j.jhydrol.2014.11.055, 2015.

Robinson, N.I., and Love, A.J.: Hidden channels of groundwater flow in Tóthian drainage basins, Advances in Water Resources 62, 71–78, http://dx.doi.org/10.1016/j.advwatres.2013.10.004, 2013.

Szijártó, M., Galsa, A., Tóth, Á., and Mádl-Szőnyi, J.: Numerical analysis of the potential for mixed thermal convection in the Buda Thermal Karst, Hungary, Journal of Hydrology: Regional Studies, 34, paper: 100783, https://doi.org/10.1016/j.ejrh.2021.100783, 2021.

Tóth, J.: A theoretical analysis of groundwater flow in small drainage basin, *Journal of Geophysical Research*, 68 (16), 4795–4812, https://doi.org/10.1029/JZ068i016p04795, 1963.

Zech, A., Zehner, B., Kolditz, O., and Attinger, S.: Impact of heterogeneous permeability distribution on the groundwater flow systems of a small sedimentary basin, *Journal of Hydrology*, 532, 90–101, http://dx.doi.org/10.1016/j.jhydrol.2015.11.030, 2016.

---

## Author Comment (AC4)

Dear Fabien Magri,

Thank you very much for your thorough review of the manuscript, your helpful comments and suggestions for further work, which have contributed to its improvement.

**General Comment**
**The manuscript by Galsa et al. presents a detailed numerical investigation into the interaction of topography-driven groundwater flow and thermohaline convection. Usually this process is referred to as "mixed convection".**
Yes, groundwater flow is considered a mixed convection system if it involves both forced convection (advection in physics) and free convection. However, we find it useful to distinguish between mixed convection regimes in which the flow is controlled by topography-driven forced convection and haline buoyancy (topohaline convection), or by topography and thermal buoyancy (topothermal convection), or even by all three components as topothermohaline convection. Thus, we can distinguish the driving forces in the umbrella term of 'mixed convection regime'.

**Specific Comments**
- **Temperature-dependent viscosity**
  Yes, the assumption of a constant dynamic viscosity of water is indeed a simplification of reality, as it was stated in the original version of the manuscript. The reason for this was to be able to investigate the dynamics of topothermohaline convection over the widest possible parameter range (water table amplitude, heat flux and salt concentration) with the available computational resources (for some models where the computation converged slowly to the quasi-stationary solution, the CPU demand of the simulation was 2–3 weeks on an Intel Server with 2 Xeon Gold 6334 CPU @ 360 GHz using 32 threads).
  On the other hand, due to the importance of the question raised, we decided to investigate the effect of the variable viscosity on the BTK model result. In the modified model, the viscosity of the water is now dependent on temperature, pressure and salt concentration (Likhachev, 2003; Adams and Bachu, 2002; Palliser and McKibbin, 1998). We found that the dynamic viscosity of the water in the parameter range under investigation is mainly influenced by temperature. Thus, in the regions of higher temperature, the viscosity of water decreased, which intensified the thermohaline convection within the confined geothermal reservoir, contributing to a decrease in the extent of the temperature anomaly and an increase in the salt flux entering through the lower boundary. The more intense flow reduced the water age in the reservoir. As the viscosity of the colder waters near the surface increased, the outflow salt flux was reduced, and the average salt concentration of the basin increased slightly. However, these changes have not altered the conceptual bigger picture established in BTK, which is that (1) topography-driven water flow is the dominant driving force in unconfined karst, while (2) thermohaline convection is the dominant flow regime in the confined reservoir, (3) the clayey Oligocene cover is saturated with saline and aged synsedimentary waters, and (4) young, cold and fresh waters from the western side are mixed with aged, hot and saline waters from the eastern side in the confined karst. (5) The water, mixed in temperature and chemical composition, continues to reach the surface in the main discharge zone along the Danube. The results of the simulation with $\mu(T,p,c)$ are presented and interpreted in the new Supplementary material attached to the manuscript, and the main conclusions are also presented in the Discussion part of the revised manuscript.

- **Boundary Conditions in Table 2**

In the synthetic simulation series, the bottom salt concentration was fixed and varied over a very wide range ($c_b$=2–200 g l$^{-1}$) in the parameter analysis to cover as many hydrogeological situations as possible, from quasi-freshwater saturated basement to evaporites (e.g. 280–345 g l$^{-1}$ for Zechstein salt in Zech et al. (2016), Kaiser et al. (2013)). The $c_b$=70 g l$^{-1}$ chosen for the base model is an 'average' value between the two extremes, which could dominate the groundwater flow in the basin for certain model parameters (e.g. low water table amplitude and heat flux) and is negligible for others (e.g. high water table amplitude and heat flux). It is therefore a suitable transition value. Reviewer 2 recommended that the ranges used in the synthetic parameter analysis should be justified and referenced, so this has been done in the revised manuscript, together with the value of the salt concentration chosen for the bottom boundary.

Unfortunately, in the case of the BTK real hydrogeological model, there is no hydrogeological/geophysical/geochemical evidence of the salinity of the basement, so we chose the value of $c_b$=35 g l$^{-1}$ to reconcile with the initial value (sedimentation in a marine environment (Mádl-Szőnyi et al., 2019)). At the same time, we also investigated how the evolution of the system is affected by the reduction of the basement salinity through diffusion. For this purpose, we defined an impermeable layer under the BTK model (between $z$=5 and 6 km asl) and imposed boundary conditions at the bottom of this layer. The simulation results are published in the Supplementary material, and the main conclusions of the results are incorporated in the Discussion section of the manuscript.

- **Boundary Condition Dominance in Figure 2**

Any boundary condition — whether it is a lateral boundary condition on the flow or a lower boundary condition on the salt concentration — affects the solution, i.e. the behaviour of the complex system. To address this question, we studied how the appearance and physical character of the thermohaline dome beneath the discharge zone are influenced by the no-flow vertical boundary condition. For this purpose, two unit basins were conjoined at the discharge zone, so the vertical boundary at the thermohaline dome was eliminated.

For the synthetic base model, the simulation was carried out in the doubled model domain, where the dome of saline, warm and aged water still formed in the middle part of the model, beneath the discharge zone, in which intense thermohaline convection was developed. The thermohaline dome, separated from the topography-driven flow regime on both sides by dynamic boundaries, was slightly weakened in its physical parameters (relative area, temperature, salt concentration and water age decreased by 4%, 9%, 12% and 21%, resp.), but the character of the flow was fully consistent with the flow/temperature/salinity/water age pattern developed in the unit half-basin. Thus, the behaviour of the topothermohaline system observed in the synthetic simulation series — thermohaline dome beneath the discharge zone — is clearly not a consequence of the lateral boundary condition. The results of the simulation are detailed in the Supplementary material, and the main conclusions are included in the Discussion section of the manuscript.

- **Implementation of Faults in BTK Model**

In this study, we have not investigated the role of the conduit/barrier faults. The faults shown in Figures 10 and 11 only separate the individual geometric units without any specific physical parameters. However, Szijártó et al. (2021) studied the role of conduit faults in the 'topothermal model' of the BTK system and found that they can influence the temperature distribution and the flow field mainly locally. If conduit faults were present in the environment of a high-temperature reservoir, they could, in theory, cause local anomalies both in temperature and salinity, but such surface manifestations do not exist in this

particular hydrogeological environment. Only at the margin of the confined and unconfined karst, along the Danube River, the occurrence of springs with different temperatures, chemical composition and yields can be seen even near each other, as already mentioned in the Discussion of the manuscript. To clarify the potential role of faults, a separate paragraph has been dedicated to this point in the Discussion.

- **BTK Model – Validation Against Observed Data:**

The primary aim of our study was to visualize the phenomenon of topothermohaline convection and to quantify the dynamics of the complex hydrogeophysical system. At the same time, we intended to demonstrate the phenomenon in a real hydrogeological field where both topography-driven forced convection and free thermohaline convection can be present. Therefore, we chose the Buda Thermal Karst system. We would like to emphasize that the study is not a case study, so — in our opinion — a comprehensive comparative quantitative analysis between the numerical model and the real system would be beyond the scope of the manuscript. These case-specific synthesising studies have already been accomplished (e.g. Mádlné Szőnyi et al, 2018; Mádl-Szőnyi et al., 2019; Mádl-Szőnyi, 2019). Therefore, we deliberately focused only on the phenomena identified within the conceptual model framework.

On the other hand, this comment of the reviewer is fully understandable, so as a compromise, in addition to the previously reported data (temperature and discharge rate of springs, water age data), we have included new data (salinity of water samples from wells) and general observations from temperature-elevation profiles in the Discussion as a validation. We are hopeful that this effort, complemented by comparison of the BTK system with other topothermohaline systems and other possible systems where topothermohaline convection may be important, will meet the reviewer's expectations.

Furthermore, minor stylistic and grammatical errors have been corrected in the revised manuscript, which — at the request of the editor — has not been uploaded at this stage of the review.

**References**

Adams, J.J., and Bachu, S.: Equations of state for basin geofluids: algorithm review and intercomparison for brines, Geofluids, 2, 257–271, https://doi.org/10.1046/j.1468-8123.2002.00041.x, 2002.

Kaiser, B.O., Cacace, M., and Scheck-Wenderoth, M.: Quaternary channels within the Northeast German Basin and their relevance on double diffusive convective transport processes: Constraints from 3-D thermohaline numerical simulations, Geochemistry, Geophysics, Geosystems, 14 (8), 3156–3175, https://agupubs.onlinelibrary.wiley.com/doi/full/10.1002/ggge.20192, 2013.

Likhachev, E.R.: Dependence of water viscosity on temperature and pressure, *Technical Physics*, 48 (4), 135–136, https://link.springer.com/content/pdf/10.1134/1.1568496.pdf, 2003.

Mádlné Szőnyi, J., Erőss, A., Havril, T., Poros, Zs., Győri, O., Tóth, Á., Csoma, A., Ronchi, P., and Mindszenty, A.: Fluids, flow systems and their mineralogical imprints in the Buda Thermal Karst, Földtani Közlöny, 148 (1), 75–96, https://doi.org/10.23928/foldt.kozl.2018.148.1.75, 2018.

Mádl-Szőnyi, J., Czauner, B., Iván, V., Tóth, Á., Simon, Sz., Erőss, A., Bodor, P., Havril, T., Boncz, L., and Sőreg, V.: Confined carbonates – Regional scale hydraulic interaction or isolation?, Marine and Petroleum Geology, 107, 591–612, https://doi.org/10.1016/j.marpetgeo.2017.06.006, 2019.

Mádl-Szőnyi, J.: Pattern of Groundwater Flow at the Boundary of Unconfined and Confined Carbonate Systems on the Example of Buda Thermal Karst and its Surroundings (in Hungarian), DSc Thesis, pp. 131, https://real-d.mtak.hu/id/eprint/1317, 2019.

Palliser, C, and McKibbin, R.: A Model for deep geothermal brines, III: Thermodynamic properties – enthalpy and viscosity, Transport in Porous Media, 33, 155–171, https://doi.org/10.1023/A:1006549810989, 1998.

Szijártó, M., Galsa, A., Tóth, Á., and Mádl-Szőnyi, J.: Numerical analysis of the potential for mixed thermal convection in the Buda Thermal Karst, Hungary, Journal of Hydrology: Regional Studies, 34, paper: 100783, https://doi.org/10.1016/j.ejrh.2021.100783, 2021.

Zech, A., Zehner, B., Kolditz, O., and Attinger, S.: Impact of heterogeneous permeability distribution on the groundwater flow systems of a small sedimentary basin, Journal of Hydrology, 532, 90–101, http://dx.doi.org/10.1016/j.jhydrol.2015.11.030, 2016.

---

## Author Comment (AC7)

**Supplementary material to**
**Topothermohaline convection – from synthetic simulations to reveal processes in a thick geothermal system**

Attila Galsa[1,2], Márk Szijártó[1,3], Ádám Tóth[4], Judit Mádl-Szőnyi[3]

[1]Department of Geophysics and Space Science, Institute of Geography and Earth Sciences, ELTE Eötvös Loránd University, Budapest 1117, Hungary
[2]Institute of Earth Physics and Space Science, HUN-REN, Sopron 9400, Hungary
[3]József and Erzsébet Tóth Endowed Hydrogeology Chair, Department of Geology, Institute of Geography and Earth Sciences, ELTE Eötvös Loránd University, Budapest 1117, Hungary
[4]Copernicus Institute of Sustainable Development, Utrecht University, Utrecht 3584, The Netherlands

*Correspondence to*: Ádám Tóth (a.z.toth@uu.nl)

**1 Water density and molecular diffusion**

In the synthetic and real case simulations, water density depended on the temperature, pressure and salt concentration as a sixth-order polynomial, a second-order polynomial and a linear function, respectively, following the formula of

$$\varrho_w(T,p,c) = \sum_{j=0}^{6} \sum_{i=0}^{2} a_{ij} p^i T^j + a_{00} \beta c, \tag{1}$$

where the first term provides the temperature- and pressure-dependent water density with an accuracy of 0.5% for a range of $p_{sat} < p \leq 100$ MPa and $0 \leq T \leq 350$ °C (Magri, 2009), and the second term is the linear effect of concentration on water density (Kohfahl et al., 2015). Coefficients for $a_{ij}$ are reported in Magri (2009), while $\beta = 7.1 \cdot 10^{-4}$ l g$^{-1}$ was used after Kohfahl et al. (2015).

A quadratic form of Arrhenius law was applied to calculate the temperature-dependent molecular diffusion coefficient,

$$D_{diff}(T) = D_0 \exp\left[\frac{A}{T^2} + \frac{B}{T} + C\right], \tag{2}$$

where $D_0 = 10^{-9}$ m$^2$ s$^{-1}$, temperature is given in K, constants of $A = -352\,450.2$ K$^2$, $B = 163.6348$ K and $C = 4.247125$ were determined by fitting on data of Easteal et al. (1989) with an error of less than 0.02%.

**2 Effect of mechanical dispersion in synthetic models**

Mechanical dispersivity in synthetic models was neglected ($\alpha_L = \alpha_T = 0$) to focus on the dynamic interaction between topography-driven forced and free thermohaline convection. Here, an additional model is presented, in which $\alpha_L = 100$ m and $\alpha_T = 10$ m were applied in the base model ($H = 50$ m, $q_{Tb} = 90$ mW m$^{-2}$, $c_b = 70$ g l$^{-1}$) to capture the effect of dispersion. Quasi-stationary solution

[Figure]

**a  Darcy flux**

$0$     $q$ [m s$^{-1}$]     $2{\cdot}10^{-8}$

**b  Temperature**

$10$     $T$ [°C]     $124$

**c  Concentration**

$0$     $c$ [g l$^{-1}$]     $70$

**d  Salinity**

fresh          brackish          saline

**e  Age**

$0$     $\tau$ [kyr]     $550$

**Figure S1 Snapshots of the (a) Darcy flux, (b) temperature, (c) salt concentration, (d) salinity zones and (e) groundwater mean age at $t$=3 Myr after the initial state in the base model (Table 2) including longitudinal ($\alpha_L$=100 m) and transverse ($\alpha_T$=10 m) dispersivity. (a) Streamlines illustrate the flow direction. (d) Salinity zones: freshwater (blue), brackish water (green) and saline water (orange).**

after 3 Myr is presented in Figure S1 showing that a more extensive thermohaline dome developed beneath the discharge zone

30  (left) compared to the base model (Fig. 2) due to the facts that (1) the transverse dispersion increased concentration flux through

the bottom boundary, and (2) the longitudinal dispersion effectively mixed the groundwaters in the basin. The phenomenon is

very similar to the behaviour of the saltwater zones in synthetic topohaline convection models, where the effect of dispersivity

was investigated in detail (Galsa et al., 2022). The relative area of the saline water zone increased from $A_{sal}$=35% to 50%

resulting in higher average concentration (from $c_{av}$=12.2 g l$^{-1}$ to 27.1 g l$^{-1}$), temperature (from $T_{av}$=34.7 °C to 45.8 °C) and

35  water age (from $\tau_{av}$=121 kyr to 170 kyr) as well as lower Darcy flux (from $q_{av}$=4.79$\cdot$10$^{-9}$ m s$^{-1}$ to 4.20 m s$^{-1}$). Thus, mechanical

dispersion, like other physical/numerical parameters (e.g. temperature- and pressure-dependent viscosity, boundary conditions), strongly influences the behaviour of the complex system.

**3 Effect of the side wall on the thermohaline dome**

Undoubtedly, all boundary conditions affect the numerical solution obtained in the model domain, whether it is a synthetic or real model, whether it refers to flow, temperature, salt concentration or water age. If a thermohaline dome is formed in a synthetic model, it always develops beneath the discharge zone, near the vertical no-flow (Darcy flux), thermally insulating (temperature) and no-flux (salt concentration and water age) boundary. Therefore, the qualitative and quantitative impact of the lateral boundary on the topothermohaline system evolving in the synthetic model was investigated, and in particular on the

[Figure]

**Figure S2 Snapshots of the (a) Darcy flux, (b) temperature, (c) salt concentration, (d) salinity zones and (e) water age at *t*=3.5 Myr after the initial state in the base model (Table 2) with doubled model domain. (a) Streamlines illustrate the flow direction. (d) Salinity zones: freshwater (blue), brackish water (green) and saline water (orange).**

thermohaline dome. In order to eliminate the side boundary effect, two sinusoidal unit basins were merged with the parameters of the base model (Table 2), in which the discharge zone was located in the middle of the domain and thus unaffected by the side boundaries and boundary conditions.

The thermohaline dome also formed under these conditions (unaffected by lateral boundaries), so it is clear that the dome beneath the discharge zone is the hydro-geophysical consequence of the system, and not an artificial phenomenon caused by the vertical boundary condition (Fig. S2). The thermohaline dome developed in the doubled model domain is now dynamically bounded on both sides, and intense thermohaline convection occurs within it. Comparing its quantitative characteristics with the base model, it was found that the relative size of the thermohaline dome decreased from $A_{sal}$=35% to 31%, and its average Darcy flux was essentially unchanged (from $q_{sal}$=2.64·10$^{-9}$ m s$^{-1}$ to 2.76·10$^{-9}$ m s$^{-1}$), its temperature, salt concentration and age decreased from $T_{sal}$=62.9 °C, $c_{sal}$=33.4 g l$^{-1}$ and $\tau_{sal}$=322 kyr to 59.2 °C, 29.3 g l$^{-1}$ and 255 kyr, respectively. Overall, it can be concluded that the thermohaline dome continues to form beneath the discharge zone, slightly weakened in its quantitative character, but the features of topothermohaline convection are consistent with the phenomena in the unit basin presented in the paper.

**4 Effect of non-constant viscosity and bottom boundary condition in the BTK model**

For simplicity and to save computational resources, the numerical calculations were performed at constant water viscosity. The present simulation illustrates the effect of temperature-, pressure- and concentration-dependent viscosity along the BTK section. The value of dynamic viscosity in the present hydrogeological model varied between $\mu$=10$^{-4}$ and 1.3·10$^{-3}$ Pa s and was mainly influenced by the temperature. In addition, an impermeable layer ($k$=0 m$^2$) of 1 km thickness was placed below the model domain, and the lower boundary conditions — which were the same as those described in Section 3.2.1 — were imposed at the bottom of this layer.

These two factors resulted in a more efficient reduction of salinity in the deep, confined reservoir because (1) the reduced viscosity due to high temperature led to more intense thermohaline convection and (2) salt was only transported by diffusion through the lower impermeable layer (Fig. S3). The lower salinity and reduced viscosity intensified the flow, thus slightly decreasing the water age. While no change in the maximum reservoir temperature was observed, its size was slightly decreased based on the model calculation. In contrast, in the regions near the discharge zone and below the Oligocene cover (HS6), a warming of a few degrees was observed, which is most likely a consequence of the more intense upwelling caused by the reduced viscosity. However, in terms of the basic characteristics of the BTK system, no substantial changes were found due to the two modified parameters (variable viscosity and modified lower boundary condition), topography-driven groundwater flow in the unconfined karst area and thermohaline convection in the confined reservoir being the dominant groundwater flow regimes. A reservoir with high geothermal potential has formed beneath the clayey cover containing saline/brackish and aged water, which can persist over geological time scales. We mention that BTK models were also calculated in which only one factor was changed (variable viscosity or bottom impermeable layer). Their solutions fell between

[Figure]

**a Darcy flux**

-13     $\log(q\,[\text{m s}^{-1}])$     -7

**b Temperature**

10     $T\,[^\circ\text{C}]$     300

**c Concentration**

0     $c\,[\text{g l}^{-1}]$     35

**d Salinity**

fresh     brackish     saline

**e Age**

0     $\tau/t\,[1]$     1

**f Viscosity**

$10^{-4}$     $\mu\,[\text{Pa s}]$     $1.3\cdot10^{-3}$

**Figure S3 Solution of the topothermohaline convection in the Buda Thermal Karst system applying temperature-, pressure and concentration-dependent viscosity and a lower impermeable layer (not shown). Distributions of (a) the Darcy flux, (b) the temperature, (c) the salt concentration, (d) the salinity zones, (e) the water age and (f) the dynamic viscosity at 1 Myr. The direction of Darcy flux is illustrated with streamlines (white). The groundwater mean age, $\tau$ is normalized by the simulation time, $t$.**

the two models presented (Fig. 10 and Fig. S3) in terms of salt concentration. The presence of the bottom impermeable layer increased the water age in the confined reservoir, while the temperature-dependent viscosity slightly decreased the temperature.

**80 References**

Magri, F.: Derivation of the coefficients of thermal expansion and compressibility for use in FEFLOW, White Papers III: DHI-WASY GmbH, Institute for Water Resource Planning and System Research, Berlin, pp. 13–23, https://hydrosoft.co.kr/sites/default/files/2024-04/Technical_Reference_FEFLOW_white_papers_vol3_ENG.pdf, 2009.

Kohfahl, C., Post, V.E.A., Hamann, E., Prommer, H., and Simmons, C.T.: Validity and slopes of the linear equation of state for natural brines in salt lake systems, Journal of Hydrology, 523, 190–195, http://dx.doi.org/10.1016/j.jhydrol.2015.01.054, 2015.

Easteal, A.J., Price, W.E., and Woolf, L.A.: Diaphragm cell for high-temperature diffusion measurements, Journal of the Chemical Society: Faraday Transactions 1, 85 (5), 1091–1097, https://doi.org/10.1039/F19898501091, 1989.

Galsa, A., Tóth, Á., Szijártó, M., Pedretti, D., and Mádl-Szőnyi, J.: Interaction of basin-scale topography- and salinity-driven groundwater flow in synthetic and real hydrogeological systems, Journal of Hydrology, 609, paper: 127695, https://doi.org/10.1016/j.jhydrol.2022.127695, 2022.